# PALI1 facilitates DNA and nucleosome binding by PRC2 and triggers an allosteric activation of catalysis

Qi Zhang[1,3], Samuel C. Agius[1,3], Sarena F. Flanigan[1], Michael Uckelmann[1], Vitalina Levina[1], Brady M. Owen[1] & Chen Davidovich [1,2 ✉]

The polycomb repressive complex 2 (PRC2) is a histone methyltransferase that maintains cell identities. JARID2 is the only accessory subunit of PRC2 that known to trigger an allosteric activation of methyltransferase. Yet, this mechanism cannot be generalised to all PRC2 variants as, in vertebrates, JARID2 is mutually exclusive with most of the accessory subunits of PRC2. Here we provide functional and structural evidence that the vertebrate-specific PRC2 accessory subunit PALI1 emerged through a convergent evolution to mimic JARID2 at the molecular level. Mechanistically, PRC2 methylates PALI1 K1241, which then binds to the PRC2-regulatory subunit EED to allosterically activate PRC2. PALI1 K1241 is methylated in mouse and human cell lines and is essential for PALI1-induced allosteric activation of PRC2. High-resolution crystal structures revealed that PALI1 mimics the regulatory interactions formed between JARID2 and EED. Independently, PALI1 also facilitates DNA and nucleosome binding by PRC2. In acute myelogenous leukemia cells, overexpression of PALI1 leads to cell differentiation, with the phenotype altered by a separation-of-function PALI1 mutation, defective in allosteric activation and active in DNA binding. Collectively, we show that PALI1 facilitates catalysis and substrate binding by PRC2 and provide evidence that subunit-induced allosteric activation is a general property of holo-PRC2 complexes.

[1] Department of Biochemistry and Molecular Biology, Biomedicine Discovery Institute, Faculty of Medicine, Nursing and Health Sciences, Monash University, Clayton, VIC, Australia. [2] EMBL-Australia and the ARC Centre of Excellence in Advanced Molecular Imaging, Clayton, VIC, Australia. [3] These authors contributed equally; Qi Zhang, Samuel C. Agius. ✉email: Chen.Davidovich@monash.edu

Over the course of evolution, gene families tend to expand[1]. Accordingly, the number of genes linked to the same function is commonly increased in vertebrates with respect to invertebrates, especially in cases of genes coding for transcriptional regulators[2,3]. For instance, the histone H3K4 methyltransferase MLL/COMPASS complex expended from one in yeast to three sub-types in fly and six complexes in human, where additional subunits emerged through each expansion[4]. Similar expansion took place for the histone ubiquitin ligase polycomb repressive complex 1 (PRC1): from two complexes in the fly to at least six variants in vertebrates[5,6]. Vertebrate-specific subunits of histone modifiers provide the opportunity to identify molecular mechanisms that are fundamental to chromatin biology and, therefore, re-emerged through the course of evolution.

The polycomb repressive complex 2 (PRC2) is a histone methyltransferase complex that is required for the maintenance of cell identity in all multicellular organisms. At the molecular level, PRC2 maintains the repressed state of developmentally expressed genes through the tri-methylation of lysine 27 in histone H3 (H3K27me3), a hallmark of facultative heterochromatin[6,7]. The core PRC2 complex includes four subunits[7–10], but it has a low histone methyltransferase activity and low affinity to DNA. Therefore, holo-PRC2 complexes include additional protein subunits—termed accessory subunits. Most of the accessory subunits of the vertebrate PRC2 emerged through gene duplication and some are vertebrate specific, with the latter poorly understood mechanistically.

The accessory subunits are collectively required for the recruitment of PRC2 to chromatin and for the regulation of its enzymatic activity[11,12]. Unbiased proteomic studies[13–17] revealed two distinct holo-PRC2 complexes—PRC2.1 and PRC2.2— defined by their mutually exclusive accessory subunits[16]. The PRC2.2 complex is nearly identical in fly and human, and includes the accessory subunits AEBP2 and JARID2. Contrarily to PRC2.2, the PRC2.1 complex went through a massive expansion over evolution: from one accessory subunit in fly, to at least five in vertebrates[13–17]. The fly PRC2.1 accompanies a single accessory subunit: Pcl. The vertebrate PRC2.1 is far more complex: it contains one of the three polycomb-like (PCL) proteins[16] (PHF1, MTF2 or PHF19) together with either EPOP[13,16] or PALI1[16,18] (also annotated as LCOR-CRA_b, LCOR isoform 3 or C10ORF12). Recent works indicate some non-redundant functions of the PRC2.1 and PRC2.2 complexes in mouse embryonic cells[11,12], but the molecular basis is unknown.

The PRC2.2-specific subunit JARID2 has two activities that were implicated in nucleating H3K27me3: chromatin binding[19,20] and allosteric stimulation of histone methyltransferase (HMTase)[21]. During JARID2-induced allosteric activation, PRC2 first di- or tri-methylates lysine 116 in JARID2 (JARID2-K116me2/3). Next, the di/tri-methyl-lysine binds to the regulatory subunit EED and triggers an allosteric activation of PRC2[21]. This mechanism is thought as a "jump-start" to activate PRC2[8]. After the nucleation of H3K27me3, histone tails carrying the H3K27me3 mark bind to the regulatory subunit EED[22] to trigger further allosteric activation of PRC2[23]. Yet, knockout of JARID2 in mouse ESC cells lacks major effect on H3K27me3 globally and locally[11,21]. This implies a parallel role taken by PRC2.1 during de novo introduction of H3K27me3, in agreement with the PRC2.1-specific subunit MTF2 being essential for this process[11,24]. Yet, a mechanism for subunit-induced allosteric activation of the PRC2.1 complex is yet to be discovered.

Multiple unbiased proteomic studies identified C10ORF12 as an accessory subunit of PRC2.1[15,16,25] and, thus, mutually exclusive with JARID2. More recently, C10ORF12 was annotated as PALI1 and identified as a vertebrate-specific protein, coded by a transcript of the *LCOR* locus[18]. Sequence homology pointed out

a paralogue of PALI1, termed PALI2, encoded by the *LCORL* locus. PALI1 is required for mouse development[18] and promotes the histone methyltransferase (HMTase) activity of PRC2 in vitro and in vivo[16,18], but the molecular mechanism is unknown.

Here, we show that PALI1 allosterically activates PRC2 and facilitates substrate binding. Mechanistically, PALI1 lysine 1241 is a substrate for PRC2 in vitro and is methylated in multiple human and mouse cell lines. Once in a di- or tri-methyl-form, PALI1-K1241me2/3 binds to the regulatory subunit EED and allosterically activates PRC2. Structural and functional evidence indicates that PALI1 has emerged through a convergent evolution to mimic the function of JARID2 within the context of the PRC2.1 complex. We also show that the PRC2-binding domain of PALI1 increases the affinity of PRC2 to DNA by >25-fold and to mononucleosome substrates by >50-fold. Allosteric activation and chromatin binding are two separate functions of PALI1, demonstrated by a separation-of-function PALI1 mutant: defective in allosteric activation but active in substrate binding. Overall, our results reveal how PRC2 is regulated by PALI1 at the molecular level, with a positive regulation at the level of both catalysis and substrate binding. More broadly, our data implies that subunit-induced allosteric activation of PRC2 is an indispensable molecular property that is permitted in most sub-types of vertebrate holo-PRC2 complexes.

## RESULTS

**PALI1 K1241 is methylated in mouse and human cell lines.** In a search for a PRC2.1 accessory subunit that could trigger an allosteric activation of PRC2, we first set out to map the PRC2 methylome in mouse and human cells. We reasoned that affinity purification of PRC2 followed by tandem mass spectrometry (AP-MS) will allow for the identification of methyl-lysines residing within assembled PRC2 complexes in vivo. Hence, we analysed multiple publicly available liquid chromatography with tandem mass spectrometry (LC-MS/MS) data originating from AP-MS experiments, where PRC2 subunits were used as baits[15,26–29] (Fig. 1a and Supplementary Data 1). Although these studies[15,26–29] were not focused on the methylation of PRC2 subunits, the high quality of the raw data allowed us to detect methyl-lysines in tryptic peptides (Supplementary Data 1). As expected, the PRC2 methylome contains the previously reported methylations in JARID2 K116[21] and EZH2 K514 and K515[30–32].

The two most frequently detected di- and tri-methyl lysines in the accessory subunits of PRC2 were in JARID2 K116 and PALI1 K1241 (Fig. 1a), with the former triggering an allosteric activation of PRC2[21]. Specifically, PALI1 K1241 is methylated in five of the seven cell lines that were tested, including both human and mouse cell lines: HEK293T (human embryonic kidney), STS26T (human malignant peripheral nerve sheath tumour), LnCAP (human prostate cancer), U2OS (human osteosarcoma) and mouse embryonic stem cells (mESC). In four of these cell lines, PALI1 K1241 was identified either in its di- or tri-methyl form (i.e. PALI1 K1241me2/3).

In some of the cell lines, we also identified methylations in PALI1 K1214 and K1219, in agreement with a previous proteomic analysis in HCT116 cells[30]. The same study also identified methylations of EZH2 K510 and K515 that have more recently been shown to regulate PRC2[31,32]. Yet, K1241 was not identified in that study[30], which used antibodies against methyl-lysines for immunoaffinity purification ahead of the mass spectrometry[30].

Hence, PALI1 and JARID2 were the only accessory subunits that were identified with di- or tri-methyl-lysine modifications that are evolutionarily conserved in mouse and human (Fig. 1a).

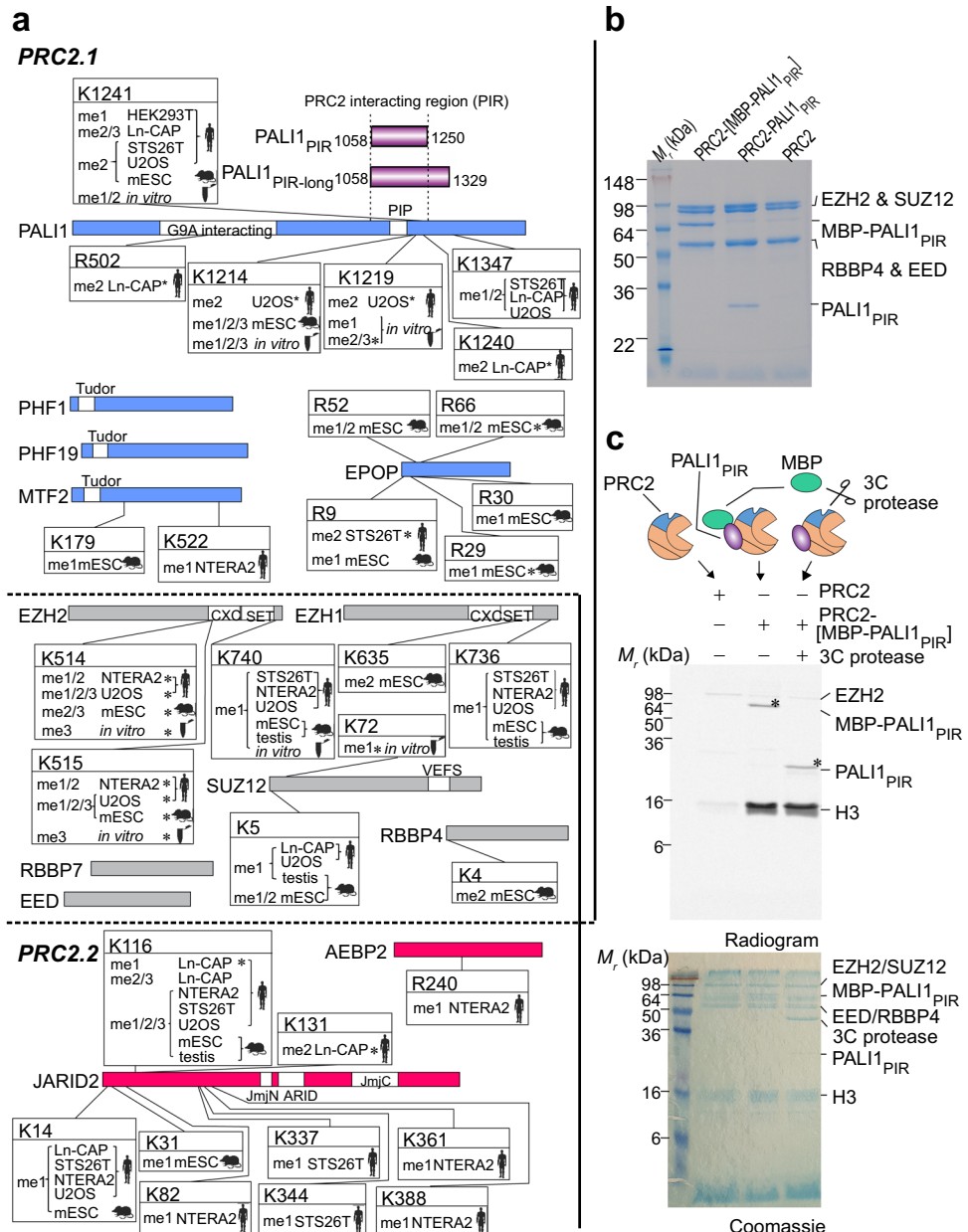

**Fig. 1 PALI1 is methylated in vitro and in vivo. a** Schematic representation of the PRC2 methylome in vivo and in vitro, as identified from MS/MS data. Mouse and human icon represent the organism of origin and cell lines are indicated (see Methods section for references and accession numbers of the raw MS/MS data). Test tube icons representing methylations in the purified recombinant human PRC2-PALI1_PIR-long. PALI truncations used in this study are indicated in purple (upper right). Residues are indicated with asterisks where the position probability of the methylation is less than 0.95 (see Supplementary Data 1 for the values). PRC2.1 and PRC2.2 accessory subunits are in blue and red, respectively, and core subunits are in grey. **b** Coomassie blue-stained SDS-PAGE of recombinant human PRC2-PALI1_PIR complexes, as indicated. **c** HTMase assay of the PRC2-[MBP-PALI1_PIR] complex using mononucleosomes substrate were carried out in the presence or absence of C3 protease to confirm that PALI1_PIR is methylated. The MBP-cleaved and uncleaved PALI1_PIR bands are indicated on the radiogram with asterisks. The HMTase assays were carried out three times with similar results and a representative gel is presented.

Di- and tri-methylated JARID2 K116 (JARID2 K116me2/3) allosterically activates the PRC2.2 complex through direct interactions with the regulatory subunit EED[21]. Based on this observation, we set out to test the hypothesis that methyl-lysines in PALI1 allosterically activate the PRC2.1 complex.

**PALI1 K1241 is methylated by PRC2 in vitro.** We next set out to determine if PALI1 can be methylated by PRC2. We expressed and purified a recombinant human PRC2 (EZH2, EED, SUZ12 and RBBP4) in a complex with the PRC2-interacting region from

PALI1 (PALI1_PIR; Fig. 1b). In vitro HMTase assay using mononucleosome substrates (Fig. 1c) confirmed that the PRC2-PALI1_PIR complex, comprising amino acids 1058–1250 from PALI1 (purple bar in Fig. 1a), is substantially more active than the core PRC2 complex. While this result is in agreement with previous reports[16,18,33], we also noted an additional band on the radiogram (Fig. 1c, marked with an asterisk). That band indicated a methylated protein that appeared only if PRC2 contained PALI1_PIR, and migrated with the apparent molecular weight of PALI1_PIR. In order to confirm that the methylated protein is

indeed PALI1_PIR, we purified a PRC2-PALI1_PIR complex with a 3C-cleavable MBP tag carried only by PALI1_PIR. We then performed the HMTase assay in the presence or absence of human rhinovirus 3C protease. The 3C-specific cleavage of the MBP-tag on PALI1_PIR led to a large shift in the migration velocity of the methylated protein, confirming it is indeed PALI1_PIR (Fig. 1c, lane 2 versus lane 3).

A similar result was obtained when we performed the same experiment using a longer truncation of PALI1 that was designed based on the previous mapping of the PRC2-interacting region from PALI1[18] (PALI1 1058-1329; termed PALI1_PIR-long herein; long purple bar in Fig. 1a). PALI1_PIR-long co-purified with PRC2 as a soluble complex (Supplementary Fig. 1a, b), albeit in multiple truncated forms. In-gel digestion with mass spectrometry subsequently identified PALI1_PIR as a more stable truncation of PALI1_PIR-long, while both constructs co-purified with PRC2 and enhanced HMTase (Fig. 1c and Supplementary Fig. 1c). These experiments confirmed that PALI1_PIR is sufficient to enhance the HMTase activity of PRC2 towards mononucleosome substrates and that PRC2 methylates PALI1_PIR in vitro. While the full-length PALI1 (PALI1_FL) was not as stable as the PALI1_PIR construct (Supplementary Fig. 1d, e), it was methylated by PRC2 and enhanced methyltransferase (Supplementary Fig. 1f). In order to identify the methylated amino acids within PALI1, we performed LC-MS/MS analysis of the recombinant PRC2-PALI1_PIR-long. As expected, we detected the previously reported methyl-lysines in EZH2 K514 and K515[31,32] (Supplementary Data 1). In PALI1_PIR-long, we identified mono- and di-methylations in K1214, K1219 and K1241 (Fig. 1a, Supplementary Data 1 and Source Data file for the MS/MS spectra), in agreement with our proteomic analysis of in vivo AP-MS data (Fig. 1a). Additionally, K1214 and K1219 were detected also in tri-methyl forms. These methylations were identified either if the complex was pre-incubated with SAM or if not, indicating that a significant fraction of the complex was purified with these modifications (Supplementary Data 1). A similar observation was previously made for EZH2 automethylation, that occurs in the recombinant protein while co-expressed with other PRC2 subunits[31]. These results confirm that three lysines within the PRC2-interacting region of PALI1 serve as a substrate for PRC2, including K1214, K1219 and K1241.

**PALI1 K1241 is required in order to enhance the HMTase activity of PRC2.** If methyl-lysines in PALI1 allosterically activate PRC2, the corresponding lysine residues are expected to be required for PALI1-mediated enhancement of HMTase. We, therefore, aimed to determine if the candidate lysine residues that we identified using MS/MS (Fig. 1a) are required or dispensable for PALI1-mediated enhancement of HMTase. We expressed and purified PRC2-PALI1_PIR and PRC2-PALI1_PIR-long mutant complexes, included all possible perturbations of the PALI1 mutations K1214A, K1219A or K1241A. Mutant complexes migrated on a gel filtration column similar to the wild type, excluding adverse effects on complex solubility (Supplementary Fig. 2b). To assess the ability of these lysine-to-alanine mutants to enhance HMTase, we carried out an in vitro HMTase assay using mononucleosomes as a substrate. The mutation K1241A in both PRC2-PALI1_PIR and PRC2-PALI1_PIR-long lead to ~50% reduction in HMTase activity, compared to the wild-type complexes (Fig. 2a and supplementary Fig. 2c, d). The two other lysine-to-alanine mutants, K1214A and K1219A, did not affect the HMTase activity of the complexes (Fig. 2a and Supplementary Fig. 2c, d). The K1241A mutant significantly reduced PALI1-mediated enhancement of HMTase in all possible perturbations that we tested, while the other two lysines, K1214 and K1219, were dispensable for methyltransferase enhancement.

Of note, while the mutation K1241A in PALI1 significantly reduced the HMTase activity of PRC2, the PRC2-PALI1_PIR K1241A mutant complex was still about 5-fold more active than the core PRC2 complex (Fig. 2a). Collectively, these data indicate that K1241 is required for complete PALI1-mediated HMTase enhancement, and implies the presence of an additional mechanism, independent of K1241 methylation (more below).

**PALI1-K1241me2/3 is sufficient in order to stimulate the HMTase activity of PRC2.** If the methylation of PALI1 K1241 is sufficient to trigger an allosteric activation of PRC2, we expected to mimic these regulatory interactions by using a short peptide, including a tri-methyl-lysine K1241 flanked by amino acids of the corresponding sequence from PALI1 (termed PALI1-K1241me3 peptide herein). Indeed, the PALI1-K1241me3 peptide significantly stimulated the HMTase activity of PRC2 towards mononucleosome substrates (Fig. 2b). Similar observations were made in the past for H3K27me3 and JARID2-K116me2/3 peptides, which allosterically activate PRC2[21,23]. We also assayed a PALI1-K1219me3 peptide, which has a smaller positive effect on the HMTase activity of PRC2. The PALI1-K1214me3 peptide was ineffective in stimulating PRC2 (Fig. 2b). Another peptide, including tri-methylations on both K1219 and K1214, exhibited only a moderate stimulation of HMTase (Fig. 2b). As expected, no stimulation of HMTase observed by unmethylated wild-type or lysine-to-arginine mutant K1241 peptides that were used as negative controls (Fig. 2c).

The level of PALI1 K1241 methylation ranges from mono- to tri-methyl-lysine in different human and mouse cell lines (Fig. 1a and Supplementary Data 1). To determine how the methylation level of PALI1 K1241 affects its ability to stimulate PRC2, we assayed the HMTase activity of the core PRC2 complex in the presence of unmethylated, mono-, di- and tri-methyl-lysine PALI1 K1241 peptides (PALI1 K1241me0-3, respectively; Fig. 2d). While the unmethylated peptide (PALI1-K1241me0) did not stimulate PRC2, any additional methyl up to the di-methyl form (PALI1-K1241me2) increased the HMTase activity of PRC2 (Fig. 2d). While the tri-methyl peptide (PALI1-K1241me3) was still efficient in HMTase stimulation, it did not increase the HMTase activity of PRC2 further beyond the di-methyl peptide (Fig. 2d). Taken together, these results indicate that PALI1-K1241me2/3 is sufficient to stimulate the HMTase activity of PRC2.

**PALI1-K1241me2/3 binds to the aromatic cage of the regulatory subunit EED to stimulate PRC2.** H3K27me3 and JARID2-K116me2/3 bind to the aromatic cage of the regulatory subunit EED to allosterically stimulate PRC2[21,23]. We, therefore, wished to determine if there is a direct link between PALI1-K1241me2/3 and EED. We first measured the affinity of EED (amino acids 40-441) for 5-carboxyfluorescein- (5-FAM, single isomer) labelled JARID2-K116me3 peptide using fluorescence anisotropy direct titrations. The JARID2-K116me3 peptide binds to EED with a dissociation constant ($K_d$) of $8.07 \pm 0.49$ μM (Fig. 3a and Supplementary Fig. 3a), consistent with previously published results[21]. Then, we quantified the $K_d$ of unlabelled peptides, using fluorescence anisotropy displacement titrations (Fig. 3a). As a positive control, we first quantified the affinity of an H3K27me3 peptide for EED, resulting with $K_d = 41.7 \pm 2.8$ μM, in agreement with a previous study[21]. The affinity of EED for PALI1-K1241me3 ($K_d = 7.49 \pm 0.59$ μM) is similar to the affinity of EED for the JARID2-K116me3 peptide ($K_d = 8.07 \pm 0.49$ μM; Fig. 3a and Supplementary Fig. 3a). We observed weak interaction between mono-methylated K1241 peptide to EED ($K_d = 241 \pm 38$ μM), while the di-methyl form increased the affinity for EED by ~10-fold ($K_d = 19.0 \pm 2.3$ μM), almost to the level of the K1241

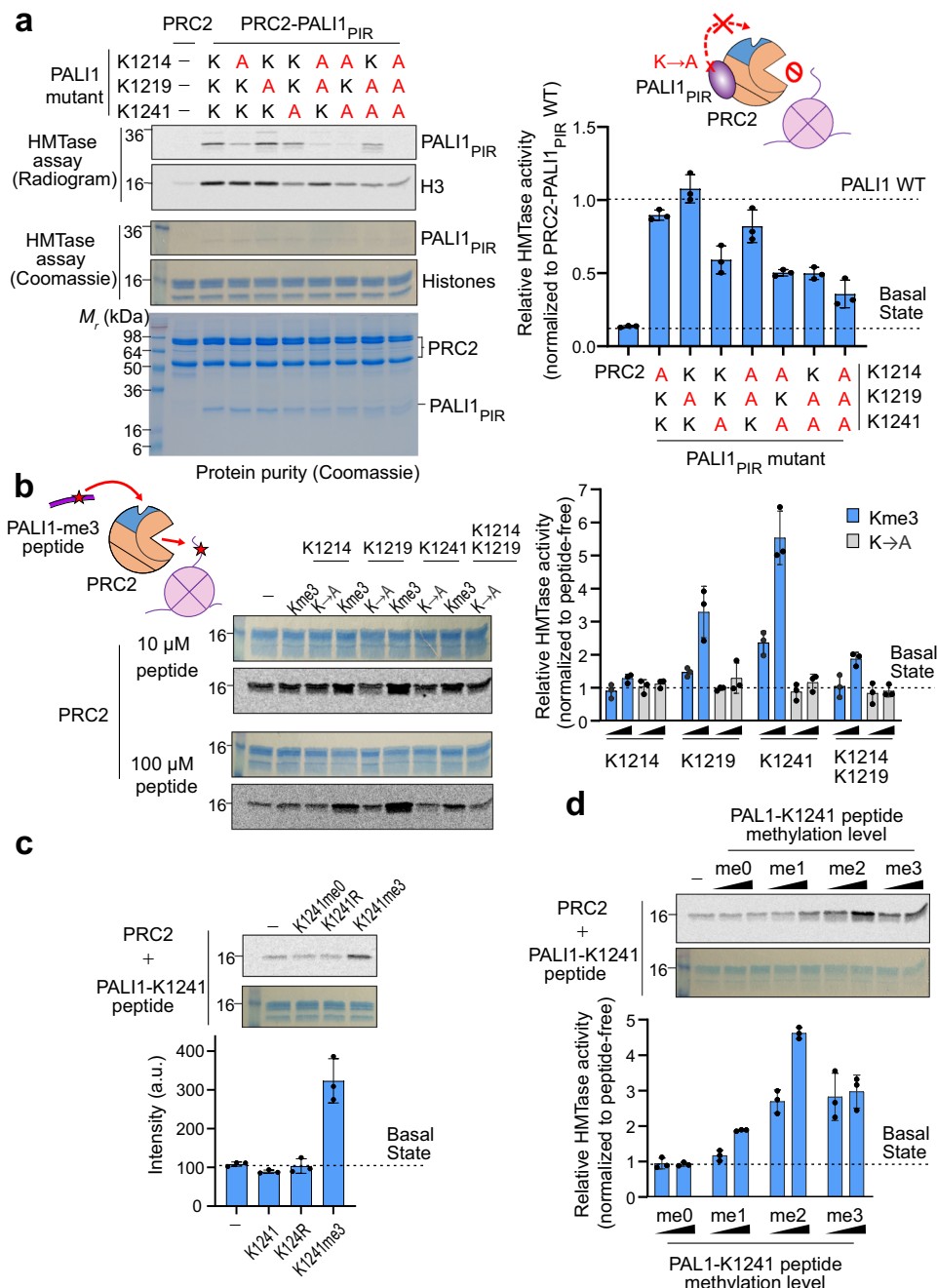

**Fig. 2 PALI1 K1241 is required and PALI1-K1241me2/3 is sufficient to stimulate the HMTase activity of PRC2. a** HMTase assays were carried out using 500 nM of wild-type or mutant recombinant complexes, as indicated, using 2 μM mononucleosomes substrate. The bar plot (right) represents mean HTMase activities, quantified using densitometry and normalised to the activity of the wild-type PRC2-PALI1$_{PIR}$. Dash lines indicate the activity of the wild-type PRC2-PALI1$_{PIR}$ (upper line) and the core PRC2 (bottom line) complexes. **b** HMTase assay performed with 500 nM PRC2, 2 μM mononucleosomes and in the presence or absence of either 10 μM or 100 μM PALI1 peptide, as indicated. The bar plot (right) represents the relative HMTase activities of PRC2 in the presence of tri-methylated (blue) or K-to-A mutated (grey) PALI1 peptides, as indicated. **c** HMTase assays of PRC2 performed as above, in the presence or absence of PALI1-K1241 peptides, as indicated. **d** HMTase assays of PRC2 performed as above, in the presence or absence of PALI1-K1241 peptides with different methylation states, as indicated. The bar plots in **b**–**d** represents the relative HMTase activities, normalised to the HMTase activity of PRC2 in its basal state (dashed line). The numbers on the left side of the gels and radiograms in this figure represent the molecular weight marker in kDa. The bar plots in all panels represent the mean of the quantification performed using densitometry over three independent replicates. Error bars shown in this Figure represent standard deviation with the observed values plotted as dots. Uncropped gel images used to generate this figure are in Supplementary Fig. 2. Source data for this figure are provided as a Source Data file.

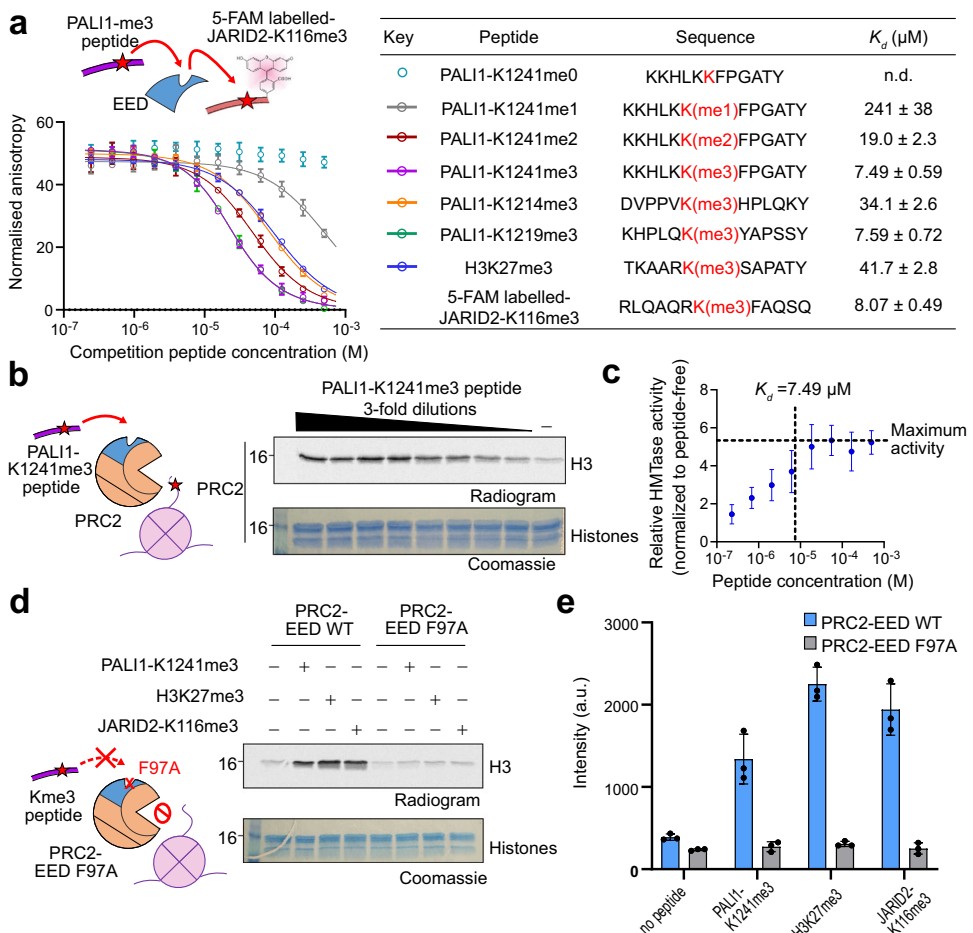

**Fig. 3 PALI1-K1241me3 binds to the aromatic cage of EED to stimulate PRC2 activity. a** Fluorescence anisotropy displacement titrations, where unlabelled peptides competed a 5-FAM labelled JARID2-K116me3 peptide (40 nM) for binding to EED (10 μM). Each data points in the plot indicate the mean of the normalised anisotropy values and the error bars represent standard deviation over three independent replicates that were carried out on different days. Dissociation constants ($K_d$) and 95% confidence bounds on the coefficient are indicated in the table. The sequence of the peptides indicated in the table, with the methyl-lysine in red. See Supplementary Fig. 3a for the binding curve of the 5-FAM labelled JARID2-K116me3 to EED. **b** HMTase assay performed using PRC2 in the presence or increased concentrtions of PALI1-K1241me3 peptides. **c** Data from **b** is presented, with the activity of the peptide-stimulated PRC2 is normalised to the activity of PRC2 in its basal state. Each data points indicate the mean of the relative HMTase activities from **b** quantified using densitometry, which was carried out on three independent replicates, and the error bars represent standard deviations. The maximum activity of PRC2 is represented by a horizontal line. The vertical dashed line represents the $K_d$ of PALI1-K1241me3 peptide and EED, as quantified in **a**. **d** HMTase assay performed using wild-type (WT) or cage-mutant PRC2 (EED F97A) in the presence or absence of stimulatory peptides, as indicated. **e** The bar plot represents the means of quantification from **d** using densitometry done on three independent replicates. Error bars represent standard deviation and the observed values indicated in dots. The numbers on the left side of the gels and radiograms in **b**, **d** represent the molecular weight marker in kDa. Uncropped gel images used to generate this figure are in Supplementary Fig. 3. Source data for this figure are provided as a Source Data file.

tri-methyl-lysine peptide ($K_d = 7.49 \pm 0.59$ μM) (Fig. 3a). In agreement with these binding assays, a titration experiment confirmed that the HMTase activity of the PRC2 core complex reached a saturation at a PALI1-K1241me3 peptide concentration of slightly above $K_d$ (Fig. 3b, c). Qualitatively, the results of these binding assays (Fig. 3a) are also in agreement with the HMTase assays done in the presence of the other peptides (Fig. 2d) and support a model where PALI1 K1241me2/3 binds to EED to stimulate the HMTase activity of PRC2.

The PALI1-K1219me3 peptide binds to EED with high affinity ($K_d = 7.59 \pm 0.72$ μM; Fig. 3a), in agreement with its ability to stimulate PRC2 (Fig. 2b). Contrarily, the PALI1-K1214me3 peptides, which did not stimulate methyltransferase (Fig. 2b), binds to EED with a 4-fold lower affinity comparing PALI1-K1219me3 and PALI1-K1241me3 (Fig. 3a).

In order to directly link between PALI1 K1241me3 and the aromatic cage of EED within the context of PRC2, we

reconstituted mutant PRC2 complex harbouring the defective cage mutation EED F97A[23]. The PALI1-K1241me3 peptide did not lead to the activation of the cage-mutant PRC2 (Fig. 3d, e), in agreement with an EED-dependent allosteric activation of PRC2. On the same line of evidence, PALI1-K1241me3-induced activation of PRC2 was inhibited by an allosteric inhibitor of PRC2, A-395, but not the negative control A-395N[34] (Supplementary Fig. 3e). Collectively, our data support a mechanism where PALI1 K1241me2/3 binds to the aromatic cage in EED to trigger an allosteric activation of PRC2.

**PALI1 and JARID2, but not H3, utilise the same interactions with the regulatory subunit EED.** Given the functional identity between PALI1-K1241me3 and -K1219me3 to JARID2-K116me3, we wished to assess for structural resemblance. We, therefore, solved the crystal structures of EED$_{76-441}$ co-crystallised with a PALI1-K1241me3 or PALI1-K1219me3 peptide (Fig. 4a and Table 1). We

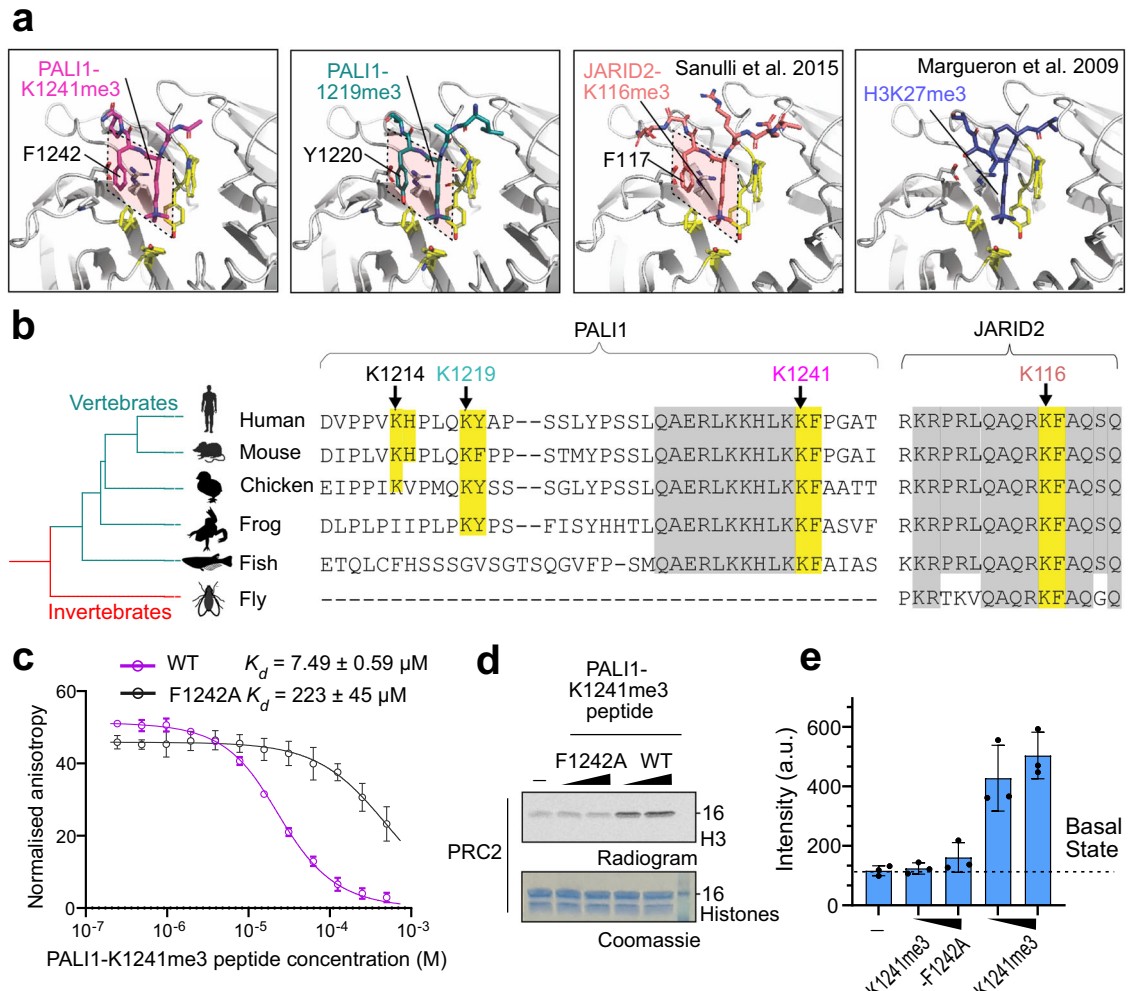

**Fig. 4 Structural basis for a convergent evolution between PALI1 to JARID2. a** High-resolution crystal structures of EED in a complex with either PALI1-K1241me3 (PDB 6V3X; this study), PALI1-K1219me3 (PDB 6V3Y; this study), JARID2-K116me3 (PDB 4X3E; Sanulli et al 2015) and EED-H3K27me3 (PDB 3IIW; Margueron et al. 2009), as indicated. The methylated lysines and their +1 adjacent conserved aromatic residues are labelled and marked in quadrilaterals. The side chains of the -1 adjacent residues to the tri-methyl-lysines—K1240 in PALI1-K1241me3 and Q1218 in PALI1-K1219me3—could not be traced and are likely disordered. The tri-methyl-lysine peptides are in sticks representation in assorted colours and EED is in grey cartoon representation, with the exception of yellow sticks that represent the aromatic cage amino acids of EED and grey sticks that represent amino acids of EED at the vicinity of the +1 conserved aromatic residue of the peptides. See Supplementary Fig. 4a for the omit electron densities of PALI1-K1241me3 and PALI1-K1219me3 peptides and Supplementary Fig. 3d, e for the sequence alignments and phylogenetic analysis. **b** Multiple sequence alignments was carried out using T-coffee[55] on entire protein sequences with the relevant regions presented (the following species were used: Homo sapiens, Mus musculus, Gallus gallus, Xenopus laevis, Oryzias latipes and Drosophila melanogaster. Phylogenetic tree was constructed based on NCBI taxonomy and visualised using iTOL[64]. The methylated lysines of JARID2 and PALI1 are labelled with arrows coloured using the same colour code as in panel **c**, with both the lysines and their adjacent aromatic residues are highlighted in yellow. Other vertebrate-conserved amino acids that are highlighted in grey. **c** Fluorescence anisotropy displacement titrations were carried out to quantify the affinity of EED to a PALI1-K1241me3 peptide in the presence or absence of the F1242A mutation. Each data points in the plot indicate the mean of the normalised anisotropy values and the error bars represent standard deviation over three independent replicates that were carried out on different days. Dissociation constants ($K_d$) and 95% confidence bounds on the coefficient are indicated. **d** HMTase assay performed using PRC2 in the presence or absence of PALI1-K1241me3 peptides with or without the F1242A mutation, as indicated. The numbers on the right side of the gel and radiogram represent the molecular weight marker in kDa. **e** The bar plot represents the means of quantification from **d** using densitometry done on three independent replicates. Error bars represent standard deviation and the observed values indicated in dots. Intensity values are in arbitrary units (a.u.). Uncropped gel images used to generate this figure are in Supplementary Fig. 4. Source data for **c**–**e** are provided as a Source Data file.

compared the resulted structures with the crystal structures of the EED-H3K27me3 and EED-JARID2-K116me3 complexes. The structures indicate that the two tri-methyl-lysine PALI1 peptides (Fig. 4a, left two panels, and Supplementary Fig. 4a) bind to EED in a conformation resembling that seen for the JARID2-K116me3 peptide (Fig. 4a, the third panel). Specifically, in all three cases the tri-methyl-lysine and its adjacent aromatic residue, in the +1 position, adopting

the same conformation when binding to EED (Fig. 4a, marked in dashed shapes). Contrarily, H3K27me3, does not have an aromatic residue at position +1, with respect to the tri-methyl-lysine, thus adopts a different binding mode to EED (Fig. 4a, right). The essential role of the phenylalanine at the +1 position has been previously demonstrated in the case of JARID2[21]. Accordingly, a F1242A mutation in the PALI1-K1241me3 peptide prevents it from binding

**Table 1 X-ray crystallography data collection and refinement statistics.**

| | PALI1-K1241me3 (PDB:6V3X) | PALI1-K1219me3 (PDB:6V3Y) |
|---|---|---|
| Data collection | | |
| Space group | $P\ 2_1 2_1 2_1$ | $P\ 2_1 2_1 2_1$ |
| Cell dimensions | | |
| $a, b, c$ (Å) | 56.2, 84.7, 90.4 | 57.8, 85.3, 91.1 |
| $\alpha, \beta, \gamma$ (°) | 90.0, 90.0, 90.0 | 90.0, 90.0, 90.0 |
| Resolution (Å) | 47.76-1.70 | 48.79-1.63 |
| $R_{sym}$ or $R_{merge}$ | 0.066 (0.631) | 0.074 (0.643) |
| $I/\sigma I$ | 16.5 (2.8) | 16.2 (3.2) |
| Completeness (%) | 97.8 (96.5) | 99.7 (97.9) |
| Redundancy | 7.6 (8.0) | 8.1 (8.1) |
| Refinement | | |
| Resolution (Å) | 35.2-1.70 | 48.8-1.63 |
| No. reflections | 47039 (4553) | 56904 (5630) |
| $R_{work}/R_{free}$ | 0.174/0.200 | 0.165/0.193 |
| No. atoms | | |
| Protein (chain A) | 2850 | 2898 |
| Ligand (chain B) | 39 | 42 |
| Water | 178 | 267 |
| $B$-factors | | |
| Protein | 22.9 | 18.0 |
| Ligand/ion | 29.1 | 24.6 |
| Water | 30.1 | 27.4 |
| R.M.S. deviations | | |
| Bond lengths (Å) | 0.006 | 0.006 |
| Bond angles (°) | 0.84 | 0.89 |
| Ramachandran plot | | |
| Favoured regions (%) | 96.6 | 96.4 |
| Allowed regions (%) | 3.4 | 3.6 |
| Disallowed regions (%) | 0.0 | 0.0 |

Values in parentheses are for the highest-resolution shell.

to EED (Fig. 4c) and from stimulating PRC2 (Fig. 4d, e). Altogether, these data and structures indicate that PALI1 and JARID2 interact with EED using their tri-methyl-lysine and its adjacent aromatic residue, despite no other sequence similarity and no common ancestor (Fig. 4b and Supplementary Fig. 4b). Of note PALI1-K1241 and the adjacent F1242, but not PALI1-K1214 or PALI1-K1219, are fully conserved across different vertebrate species (Fig. 4b). This implies a biological significance of PALI1-K1241me2/3 in regulating PRC2 across vertebrates.

**PALI2 K1558 mimics PALI1 K1241 in the allosteric regulation of PRC2 in vitro.** PALI1 is mutually exclusive with EPOP for PRC2 binding[13,16] and share high sequence similarity with PALI2[18]. We did not identify methyl lysines in EPOP (Fig. 1a) and have no factual basis to consider it as an allosteric regulator of PRC2, in agreement with the role of EPOP as a negative regulator of PRC2 in cells[13,35]. When it comes to PALI2, sequence alignment revealed that PALI2 K1558 and its adjacent phenylalanine are identical to the corresponding amino acids in PALI1: PALI1 K1241 and its adjacent phenylalanine (Fig. 5a). Accordingly, a PALI2-K1558me3 peptide, but not PALI1-K1558 (unmethylated), binds to EED with high affinity ($K_d = 8.81 \pm 1.48\ \mu M$, Fig. 5b) and stimulates the enzymatic activity of PRC2 (Fig. 5c). Next, we reconstituted the recombinant PRC2-PALI2$_{1330-1641}$ complex, with the PALI2 construct designed based on its homology to PALI1$_{PIR}$. Despite being less stable than PALI1$_{PIR}$ (Supplementary Fig. 5b, c), the recombinant PRC2-PALI2$_{1330-1641}$ complex exhibited high enzymatic activity

compared to PRC2 (Fig. 5d) and a small but notable level of PALI2 methylation observed (Supplementary Fig. 5d). Altogether, these results indicate that PALI2 may regulate PRC2 in a similar mechanism as identified above for PALI1. It is worth noting that PALI2 was not detected as an interacting partner of PRC2 in most of the AP-MS dataset that we analysed, with the only two exceptions where in human U2OS and mouse testis, where in both cases PALI2 detected with a poor coverage (0.6% and 6.3–12.8%, respectively). The poor coverage of PALI2 in these datasets were not allowed identifying its methylation. While it remains to be determined under what circumstances PALI2 regulates PRC2 in vivo, our data indicates that it has the capacity to bind PRC2 and to trigger an allosteric activation.

**PALI1 facilitates DNA binding by PRC2, with allosteric activation being dispensable for this function.** The triple mutant complex, PRC2-PALI1$_{PIR-long}$ K1241A, K1219A and K1241A, is defective in allosterically stimulating HMTase but was still more active than the core PRC2 complex (Fig. 2a and supplementary Fig. 2c, d). This mutant complex cannot harbour any of the methylations that we identified in PALI1 (Fig. 1a). Hence, we suspected that the PRC2-interacting region within PALI1 regulates HMTase in an additional mechanism. In vitro HMTase assays previously demonstrated that the HMTase activity of PRC2 is enhanced by several of its DNA-binding accessory subunits, including MTF2, PHF19 and AEBP2[36]. We therefore wished to determine if the PRC2-interacting region of PALI1 increases the affinity of PRC2 to DNA.

To directly test if the PRC2-interacting region of PALI1 can facilitate DNA binding, we first set out to quantify the affinity of PRC2-PALI1$_{PIR}$ to DNA using fluorescence anisotropy. We used a DNA probe designed to mimic 46 bases long dsDNA from a CpG island of the *CDKN2B* gene (termed CpG46 DNA, see Supplementary Table 1 for the DNA sequence). The affinity of the PRC2-PALI1$_{PIR}$ to CpG46 DNA ($K_d = 155 \pm 26\ nM$) was >20-fold higher than the affinity to the PRC2 core complex to the same DNA probe ($K_d > 4\ \mu M$) (Fig. 6a), indicating that PALI1 facilitates DNA binding. Accordingly, Chromatin immunoprecipitation (ChIP) with qPCR confirmed that ectopically expressed PALI1 binds to chromatin in cells (Supplementary Fig. 10), in agreement with a previous study that identified PALI1 in the chromatin-bound nuclear fraction[18]. In vitro, PALI1$_{PIR}$ binds DNA even in the absence of PRC2, although with a reduced affinity ($K_d = 2.3 \pm 0.3\ \mu M$; Supplementary Fig. 6c). The PALI1$_{PIR}$ construct can enhance the HMTase activity of the core PRC2 even if they were not co-expressed together but rather combined just before the reaction (Supplementary Fig. 2h).

The DNA-binding activity of PALI1 was not specific to the CpG46 DNA probe: PRC2-PALI1$_{PIR}$ binds to DNA tightly even after the DNA probe was mutated to disrupt all the CpG sequences and to reduce the GC content from 79 to 21% (CpG46 mt DNA, see Methods section for DNA sequence), with $K_d = 73.7 \pm 10.0\ nM$ for CpG46 mt DNA (Fig. 6a). Conversely, PALI1$_{PIR}$ did not significantly increase the affinity of PRC2 to a G-tract RNA (Supplementary Fig. 6a), which interacts with core PRC2 subunits[36,37]. Accordingly, this data indicates that the PRC2-interacting region of PALI1 facilitates high-affinity interactions between PRC2 and DNA, not RNA, without an apparent DNA-sequence selectivity. Some level of target specificity might be achieved in cells by additional factors. For instance, PCL proteins were implicated in binding to CpG islands and can bind to PRC2 together with PALI1[16,18].

**PALI1 facilitates nucleosome binding by PRC2.** Given that PALI1 facilitates DNA binding (Fig. 6a), we next wished to

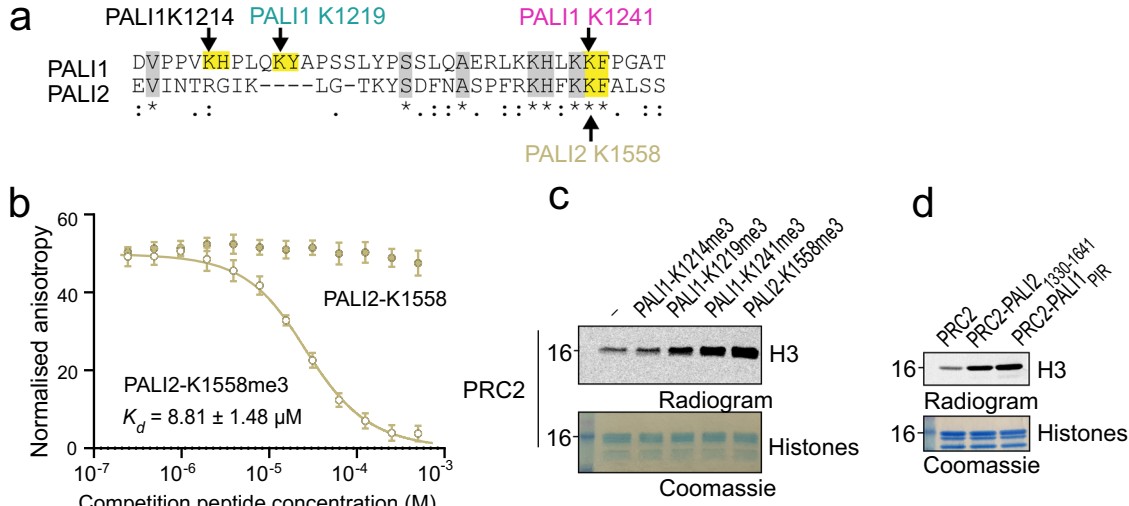

**Fig. 5 PALI2-K1558me3 mimics PALI1-K1241me3 in the allosteric activation of PRC2 in vitro. a** Sequence alignment of PALI1 and PALI2 around the methylated lysine region. The methylated lysines of PALI1 and the putative methylated lysine in PALI2 are labelled with arrows coloured using the same colour code as in panel **c** of Fig. 4. The methylated lysines and their adjacent aromatic residues are highlighted in yellow. Other conserved amino acids are highlighted in grey. **b** Fluorescence anisotropy displacement titrations were used to quantify the binding of PALI2-K1558 peptides to EED. Each data points in the plot indicate the mean of the normalised anisotropy values and the error bars represent standard deviation over three independent replicates that were carried out on different days. Dissociation constants ($K_d$) and 95% confidence bounds on the coefficient are indicated. **c** HMTase assay performed using PRC2 in the presence or absence of stimulatory peptides, as indicated. **d** HMTase assay performed using recombinant complexes, as indicated. HMTase activities in **c**, **d** were repeated three times. The numbers on the left side of the gels and radiograms in **c**, **d** represent the molecular weight marker in kDa. Uncropped gel images used to generate this figure are in Supplementary Fig. 5. Source data for **b** are provided as a Source Data file.

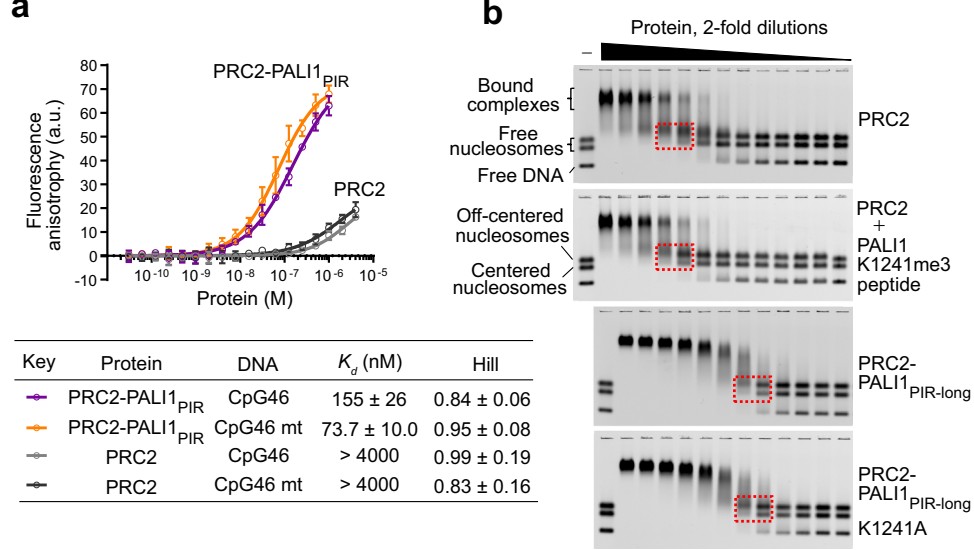

**Fig. 6 PALI1 facilitates DNA binding by PRC2. a** Fluorescence anisotropy used to quantify the affinity of PRC2 complexes to fluorescein-labelled CpG46 or CpG46 mt DNA. Data represent the mean of three independent experiments that were carried out on different days and error bars represent standard deviation. Dissociation constants ($K_d$) and Hill coefficients are indicated in the table, including their standard error. Fluorescence anisotropy values are in arbitrary units (a.u.). **b** EMSA used to quantify the affinity of the indicated PRC2 complexes for a mixture of Cy5-labelled mononucleosomes and free DNA of the same sequence. Dashed boxes indicate the mononucleosome bands near the $K_d$ concentration of the protein, where half of the labelled mononucleosomes are shifted (for quantification, see Supplementary Fig. 6b, c). Source data for **a** are provided as a Source Data file.

determine if PALI1 could facilitate substrate binding. We performed electrophoretic mobility shift assays (EMSA) using a fluorescently labelled mononucleosome probe that was reconstituted using a 182-bp-long DNA. The probe was reconstituted to allow for the simultaneous detection of the interactions between PRC2 and nucleosomes, at either the centred or the off-centred positions[38], or between PRC2 and the free DNA (Fig. 6b).

In agreement with a previous work[39], the PRC2 core complex exhibited moderate affinity for mononucleosomes ($K_d = 330 \pm 30$ nM). Remarkably, the PRC2$_{PIR-long}$ construct increased the affinity of PRC2 to nucleosomes by >15-fold ($K_d = 19.0 \pm 0.6$ nM) compared to the PRC2 core complex (Supplementary Fig. 6b). To determine if PALI1 enhances substrate binding in a mechanism linked to allosteric activation, we quantified the affinity of the

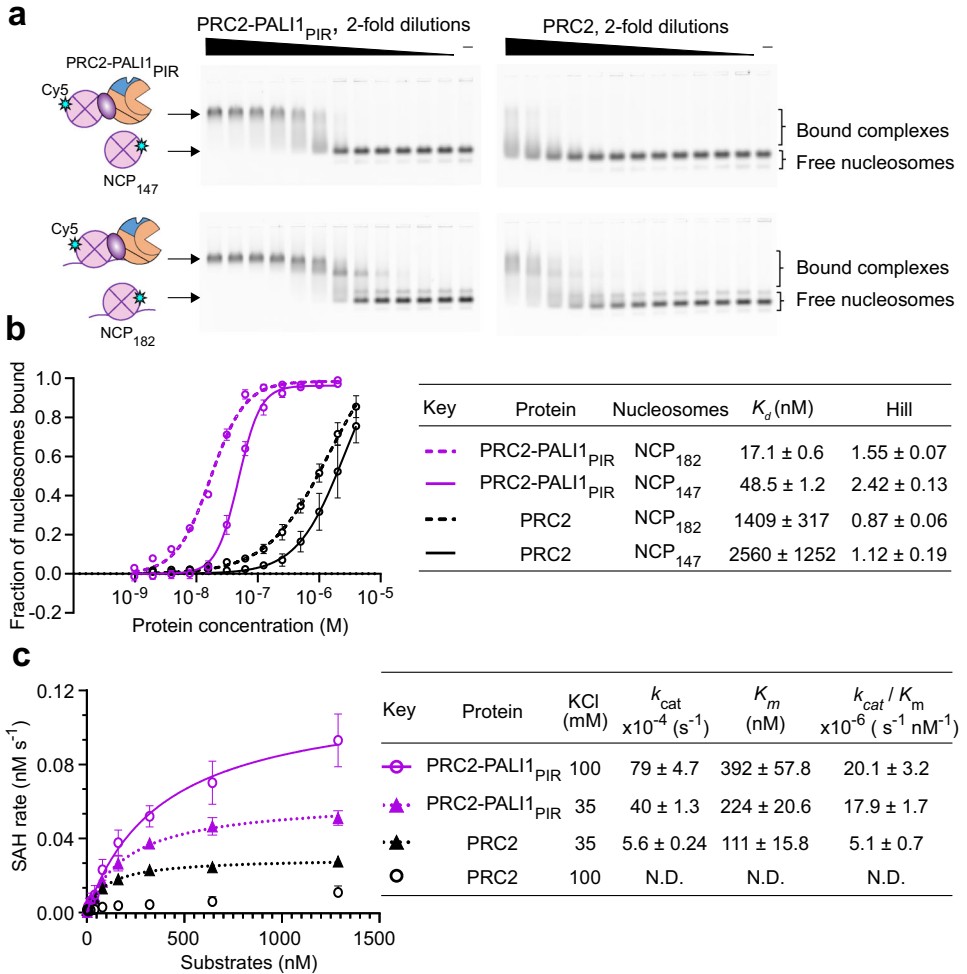

**Fig. 7 PALI1 facilitates nucleosome binding by PRC2. a** EMSA used to quantify the affinity of the indicated PRC2 complexes for H2A-Cy5-labelled nucleosome core particles substrate (top, $NCP_{147}$) and mononucleosomes with 35 bp linker DNA substrate (bottom, $NCP_{182}$). In all, 2-fold dilutions of the protein were carried out, starting from 2 μM PRC2-PALI1$_{PIR}$ and 4 μM PRC2. **b** Quantification of the EMSA from **a**. Data represent the mean of three independent experiments and the error bars represent standard deviations. Standard errors of dissociation constants ($K_d$) and Hill coefficients are indicated in the table. **c** Michaelis–Menten kinetic analysis of PRC2-PALI1$_{PIR}$ (15 nM) and PRC2 complex (50 nM) on a nucleosomal array substrate. Colour key for the plot and KCl concentrations are indicated in the table. Data represents the mean of three independent experiments that were carried out on different days and error bars represent standard deviation. $k_{cat}$, $K_M$, and the catalytic efficiency ($k_{cat}/K_M$) values are indicated with standard errors. Substrate concentrations were defined by the octamer concentrations in the arrays. Progress curves confirmed that the reaction is at the linear range and evidence for the substrate quality are in Supplementary Fig. 7. Source data for this figure are provided as a Source Data file.

PRC2 core complex to nucleosomes after pre-incubation with a PALI1-K1241me3 peptide (Fig. 6b and Supplementary Fig. 6b). The PALI1-K1241me3 peptide did not increase the affinity of the PRC2 core to nucleosomes (Supplementary Fig. 6b), thus suggesting that PALI1 facilitates an allosteric activation and substrate binding by independent mechanisms. Accordingly, the PRC2-PALI1$_{PIR-long}$ K1241A mutant, which is defective in the allosteric activation (Supplementary Fig. 2c, d), binds nucleosomes with a similar affinity to the wild-type PRC2-PALI1$_{PIR-long}$ (<2-fold $K_d$) (Fig. 6b and Supplementary Fig. 6b, c). Collectively, these results indicate that the PRC2-binding domain of PALI1 facilitates DNA and substrate binding in addition to—and independent of—allosteric activation.

To directly examine the contribution of PALI1 to the nucleosome binding activity of PRC2, we carried out EMSA using Cy5-labelled nucleosome core particles with ($NCP_{182}$) or without ($NCP_{147}$) a linker DNA (Fig. 7a). Strikingly, PALI1$_{PIR}$ increased the affinity of PRC2 to $NCP_{147}$ by >50-fold (Fig. 7b; $K_d$ = 2560 ± 1250 nM for PRC2 and $K_d$ = 48.5 ± 1.2 for PRC2-PALI1$_{PIR}$). This indicates that a DNA linker is dispensable for

high-affinity interactions between PRC2-PALI1$_{PIR}$ to a nucleosome. Adding a DNA linker slightly increased the affinity of PRC2-PALI1$_{PIR}$ to the nucleosome constructs (Fig. 7b; $K_d$ = 17.1 ± 0.6 nM for PRC2-PALI1$_{PIR}$, when using the a $NCP_{182}$ probe), in agreement with our observation that PALI1 facilitates DNA binding (Fig. 6).

Given the high affinity of PALI1 to DNA (Fig. 6) and nucleosomes (Fig. 7), we anticipated that it would facilitate the chromatin-binding activity of PRC2 during histone methyltransferase. To test this, we next carried out quantitative histone methyltransferase assays (Fig. 7c) using a nucleosome array. In agreement with the function of PALI1 in allosterically activating PRC2 (Figs. 2–4), $k_{cat}$ of PRC2-PALI1$_{PIR}$ was over 7-fold higher than that of the core complex (Fig. 7c). Most importantly, $K_M$ values of the PRC2-PALI1$_{PIR}$ complex were almost insensitive to the salt concentration, with <2-fold increment in $K_M$ while comparing low salt to high salt reaction buffers ($K_M$ = 224 ± 20.6 nM in 35 mM KCl and 392 ± 57.8 nM in 100 mM KCl; Fig. 7c). Accordingly, the catalytic efficiency of PRC2-PALI1$_{PIR}$ was almost the same in high or low salt buffers (1.1-fold change,

see Fig. 7c for $k_{cat}/K_M$ values). This suggests that the electrostatic potential is not the key driver for the interactions between PRC2-PALI1$_{PIR}$ to chromatin during methyltransferase. Contrarily, the PRC2 core complex was very sensitive to the salt concentration, where increasing the KCl concentration from 35 mM to 100 mM significantly reduced its activity (Fig. 7). Collectively, our qualitative histone methyltransferase assays (Figs. 2–4), quantitative histone methyltransferase kinetic assays (Fig. 7c), the nucleosome binding assays (Fig. 7a, b) and the DNA binding assays (Fig. 6) support a mechanism where the PRC2-binding domain of PALI1 allosterically activates PRC2 while anchoring it on chromatin through interactions with DNA and nucleosomes to ensure a close proximity to the histone tail substrates.

**Overexpression of PALI1 triggers cell differentiation in chronic myeloid leukaemia cells, with the phenotype altered by an allosteric-defective mutant**. One paper reported a large increment of global H3K27me3 in HeLa cells after C10ORF12 overexpression[33]. Yet, we did not detect a significant change of global H3K27me3 while overexpressing PALI1 in K562 (Supplementary Fig. 8a), HEK293T (Supplementary Fig. 8b, left) and HeLa (Supplementary Fig. 8b, right) cells. Little is known about the cellular function of PALI1, but our data thus far indicates a resemblance to JARID2 at the molecular level. JARID2 is frequently deleted in the leukaemic transformation of chronic myeloid malignancies[40]. Accordingly, the overexpression of JARID2 leads to reduced proliferation in leukaemia cell lines and it has been proposed to serve as a tumour suppressor in leukaemia[41]. Deletion of the LCOR locus has been reported in B-cell acute lymphoblastic leukaemia[42]. Given the functional resemblance between PALI1 to JARID2 and their potential role as tumour suppressors in hematopoietic malignancies, we wished to determine if PALI1 has a negative effect on the proliferation of myelogenous leukaemia cells. Indeed, competitive proliferation assay indicated that the overexpression of PALI1, but not the negative control LacZ, in a human chronic myeloid leukaemia cell line (K562) leads to a strong reduction in cell proliferation (Fig. 8a, b).

In addition to reduced proliferation, we noticed that the overexpression of PALI1, but not the negative control LacZ, in K562 cells led to the pelleted cells becoming red in colour (Fig. 8c). This observation suggested differentiation along the erythroid lineage[43], in accord with the reduced cell proliferation (Fig. 8a, b). We therefore set out to detect the erythroid differentiation marker CD235a, which increases during erythropoiesis[44], and the erythroid precursor marker CD44 that decreases during erythroid differentiation[45] (see illustration at Fig. 8e). Indeed, the overexpression of PALI1, but not LacZ, led to increased expression of CD235a and reduction of CD44, in accord with differentiation along the erythroid lineage (Fig. 8d, e and Supplementary Fig. 9a, b).

We next assayed the PALI1 K1241A separation-of-function mutant, which is defective in allosteric activation (Fig. 2 and Supplementary Fig. 2c, d) and active in DNA binding (Fig. 6b and Supplementary Fig. 6b). The red colour of the cells was comparable either if the PALI1 wild-type or K1241A mutant was expressed (Fig. 8c, d), and both the mutant and the wild-type PALI1 significantly reduced K562 cell proliferation (Fig. 8b). Yet, cells expressing the PALI1 K1241A mutant proliferated faster than cells expressing the wild-type protein (Fig. 8b) and exhibited different expression levels of the erythroid markers CD235a and CD44 (Fig. 8c, d and Supplementary Fig. 9a, b). This result was insensitive to variations of expression level of the different constructs (Supplementary Fig. 9c). This is possibly because the protein level of PALI1 in the cell is limited by the number of

PRC2 molecules, indicative by the depletion of ectopically expressed PALI1 upon doxycycline-induced knockout of EED (Supplementary Fig. 9f). The changes in CD expression levels in K562 cells are not necessarily a direct effect of PALI1 overexpression. Yet, these effects are dependent on EED (Supplementary Fig. 9d, e) and therefore are likely reporting for the ability of the PALI1 wild-type and mutant to regulate transcription in these cells.

The importance of PALI1 as a positive regulator of PRC2 has been previously shown through knockout experiments that led to embryonic lethality in the mouse and a reduction in the H3K27me3 mark at a portion of PRC2-target genes in mouse embryonic cells[18]. Data herein (Figs. 1–7) decipher the molecular mechanism of PALI1-mediated regulation of PRC2: PALI1 mechanistically resembles JARID2 to allosterically activate PRC2 and, independently, facilitates DNA and nucleosome binding (Fig. 9).

**Discussion**

Our data indicate that the PRC2-interacting domain of PALI1 is sufficient to enhance the HMTase activity of PRC2 by two independent mechanisms: (i) allosteric activation of catalysis (Fig. 2) and (ii) DNA binding (Figs. 6 and 7). Hence, as little as 193 amino acids in PALI1 (1058–1250) are sufficient to bind PRC2, bind DNA and stimulate methyltransferase, with the other >1500 amino acids in PALI1 likely available to engage in other tasks.

**A convergent evolution between PALI1 and JARID2**. The nucleation of H3K27me3 de novo takes place when transcription programmes are changed during cell differentiation and newly repressed genes acquire the H3K27me3 mark. In the context of the PRC2.2, JARID2 facilitates the nucleation of H3K27me3[24], aided by its chromatin-binding activity[19,20,46] and its ability to allosterically stimulate PRC2[21]. In the context of PRC2.1, MTF2 functions in the nucleation of H3K27me3[24], with a proposed contribution to its DNA-binding activity[47,48]. Although MTF2 does not allosterically stimulate PRC2, it coexists in the PRC2.1 complex with PALI1[16,18,25]. Through the discovery that PALI1 allosterically activates PRC2 (Figs. 2 and 3) and facilitates substrate binding (Figs. 6 and 7), we reveal a striking functional resemblance between PALI1 to JARID2: (i) Both PALI1 and JARID2 bind to nucleosomes: the former likely with the aid of its DNA binding activity (Fig. 6) and the latter through interactions with H2AK119-ubiquitinated chromatin[19,20]. (ii) JARID2 comprises the PRC2.2 complex together with the chromatin-binding subunit AEBP2 while PALI1 binds to the PRC2.1 complex together with a polycomb-like DNA binding subunit (PHF1, MTF2 or PHF19)[16,18,25]. (iii) PRC2 methylates PALI1 and JARID2, with the di- or tri-lysine then binds to EED for allosteric activation of PRC2 (Figs. 2 and 3 and Sanulli et al.[21]). (iv) For their interactions with EED, both PALI1 and JARID2 using an aromatic residue, located at the +1 position with respect to the methylated lysine (Fig. 4a). The importance of the +1 adjacent aromatic residue supported by the JARID2 F117A mutation that prevented both EED binding and the stimulation of PRC2[21] and data herein using the PALI1 F1242A mutant peptide (Fig. 4c–e).

Strictly, despite these mechanistic and structural similarities, PALI1 and JARID2 have no common ancestor: PALI1 is a vertebrate-specific protein[18] and JARID2 is conserved in fly and human (Fig. 4b). Therefore, we propose that PALI1 has emerged in vertebrates as the result of convergent evolution, under a selection pressure to mimic some of the molecular functions of JARID2 within the context of the PRC2.1 complex.

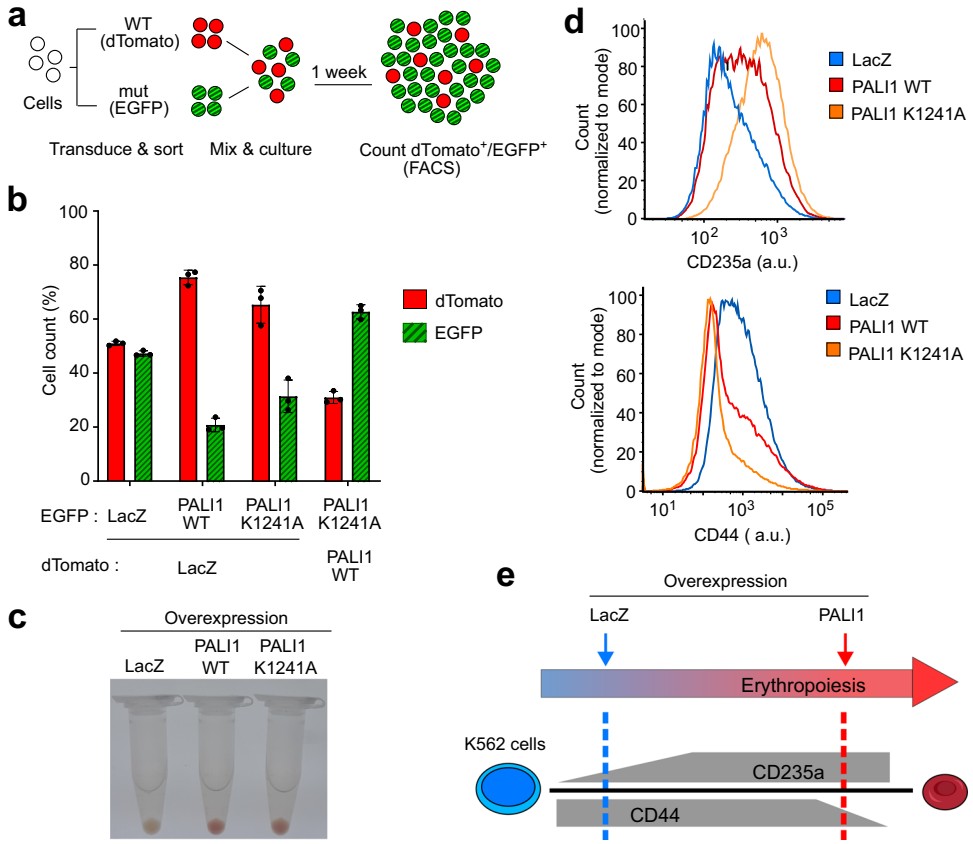

**Fig. 8 Overexpression of PALI1 triggers K562 cell differentiation along the erythroid lineage, with the effect altered by a separation-of-function allosteric-defective PALI1 mutant (K1241A). a** Schematic illustration of the competitive cell proliferation assay used to directly compare the proliferation of cells transduced with different constructs. Cells overexpressing different proteins and fluorophores are co-cultured, with the higher-proliferating cells outcompete the lower-proliferating cells. At the end of the experiment, the number of cells carries each vector is quantified using flow cytometry. **b** Competitive cell proliferation assay of human chronic myeloid leukaemia (K562) cells overexpressing either PALI1 wild-type (WT), the separation-of-function PALI1 mutant (K1241A) or the LacZ negative control, as indicated. The bar plot represents the mean percentage of each cell population after 7 days of competition, as quantified using flow cytometry. Error bars represent standard deviations derived from three independent experiments carried out on different days with the observed values indicated by dots. Evidence for protein expression and nuclear localisation are in Supplementary Fig. 5a. Source data are provided as a Source Data file. **c** Colour photographs of pelleted K562 cells overexpressing different proteins: LacZ (left), PALI1 WT (middle) and PALI1-K1241A mutant (right). **d** Representative histograms generated from flow cytometric analysis of the differentiation markers CD235a (top) and CD44 (bottom) exhibited by K562 cells overexpressing PALI1 WT (red), PALI1 K1241A mutant (orange) or LacZ (blue). The other two replicates shown in Supplementary Fig. 9a, b and evidence for protein expression are in Supplementary Fig 8a. **e** Illustration of the expected expression level of CD235a and CD44 during erythropoiesis (grey) and the actual observed values represented in dashed lines coloured in red and blue for PALI1 and the negative control LacZ, respectively.

**PALI1 provides PRC2 with means to gauge its own enzymatic activity before adding a stimulus.** EZH2 automethylation is proposed to modulate the HMTase activity of PRC2 in response to molecular cues, including the presence of histone H3 tails and SAM concentration[31]. A similar principle might apply for PALI1. This mechanism seems to affect mainly the methylation of histone substrates, not the automethylation of EZH2. Specifically, a similar level of EZH2 methylation observed in the presence of the methyl-defective mutants PALI1 K1214A, K1219A and K1241A or the presence of PALI1-methyl peptides (Supplementary Fig. 2c, e). Future studies are still needed in order to identify whether PALI1 (and JARID2) are methylated in cis or by another PRC2 complex and if the degree of methylation changes during normal development or in cancer and other pathologies. Yet, our analysis indicates that PRC2 has the capacity to methylate PALI1 (Fig. 1b, c) and that the degree of PALI1 K1241 methylation varies between cell lines (Fig. 1a). Our data also indicates that PRC2-induced methylation of PALI1 in vitro is more efficient in the presence of nucleosome substrates (Supplementary Fig. 2h).

Given these observations, it is plausible that PALI1 provides PRC2 with means to gauge its own enzymatic activity before applying an additional stimulus.

**PALI1 as a potential bridge between the H3K27me3 and H3K9me2 repressive marks.** Amino acids in regions of PALI1 dispensable for the regulation of PRC2 bind to the H3K9-methyltransferase G9a[18]. PRC2 and G9a share a portion of their genomic targets[49] and are physically associated[49,50]. In ES cells, G9a contributes to H3K27 methylation in vivo, with the global H3K27me1—but not H3K27me2/3—reduced upon knockout of G9a[51]. Affinity purification mass spectrometry (AP-MS) with either C10ORF12[25] or PALI1[18] used as baits detected the subunits of the G9a–GLP H3K9me1/2 methyltransferase complex, including G9a, GLP and WIZ. Importantly, the PRC2-interacting domain of PALI1 is distinct to its G9a-interacting region[18]. Our data indicate that the PRC2-binding region of PALI1 is sufficient (i) to bind to PRC2 (Fig. 1), (ii) to promote DNA and nucleosome

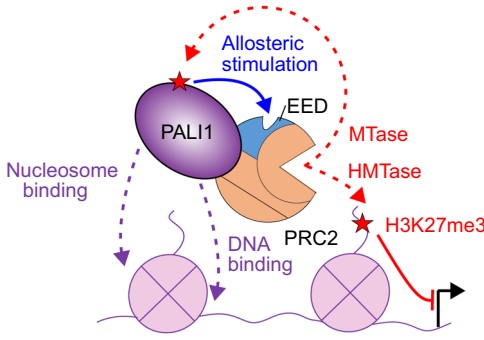

**Fig. 9 A model for PALI1-mediated regulation of PRC2.** PALI1 regulates PRC2 through two independent mechanisms: (i) PRC2 methylates PALI1 K1241 and possibly K1219, which then binds to the regulatory subunit EED to trigger an allosteric activation of PRC2. (ii) PALI1 facilitates DNA binding and nucleosome substrate binding. PALI1 is in deep purple, EED in light blue and other PRC2 core subunits are in orange. Dashed red arrows represent methylation, with the red stars represent methyl-lysines. The blue arrow represents a positive regulation by an allosteric activation, the dashed purple arrow represents DNA and nucleosome binding and the continues red line represents a transcriptional repression.

binding (Figs. 6 and 7) and (iii) to trigger an allosteric stimulation of catalysis (Fig. 2): three prerequisites for the nucleation of H3K27me3 de novo[21]. Our data also indicate that PRC2 is required for the stability of PALI1 in cells (Supplementary Fig. 9f). This indirectly implies that G9a would have to "share" its PALI1-binding partner with PRC2, as otherwise PALI1 is unstable. While experiments in cells are required in order to determine if PALI1 could nucleate both H3K27me3 and H3K9me2 de novo during cell differentiation, it does have the molecular characteristics for that.

Recent studies showed that PRC2.1 and PRC2.2 synergise and share most of their target genes[11], with their accessory subunits collectively required[12]. While a previous attempt to identify the binding sites of PALI1 on chromatin using ChIP was reported as unsuccessful[18], the PRC2.1 complex is localised at a minority of unique genomic sites, independently of PRC2.2[11]. We measured small variation between the affinity of the PRC2-PALI1$_{PIR}$ complex for the CpG-reach DNA (CpG46 $K_d = 155 \pm 26$ nM) comparing a size-matched DNA lacking CpG sequences (CpG46 mt; $K_d = 73.7 \pm 10$ nM) (Fig. 6a). It is, therefore, possible that PALI1 could allow for some degree of target specificity, utilising some variations in affinities for DNA combined with context-specific chromatin binding. Such context-specific binding could be attributed to interactions with the G9a complex and the CTBP proteins binding to the N-terminal domain of PALI1[18]. This model is in agreement with the view that a combination of factors and interactions are responsible for the recruitment of PRC2 to its target genes[7].

Collectively, our data indicate that the PRC2-binding domain of PALI1 enhances H3K27-methyltransferase by two independent mechanisms (Fig. 9): (i) DNA- and nucleosome-binding and (ii) allosteric stimulation. The remarkable mechanistic resemblance between PALI1 and JARID2 indicates convergent evolution of the regulation of the PRC2.1 and PRC2.2 complexes, respectively. More broadly, it implies that subunit-induced allosteric activation is an indispensable property of a holo-PRC2 complex in vertebrates.

## Methods
**Protein expression and purification.** The cloning of constructs for the expression of the full-length sequences encoding for human EZH2, SUZ12, RBBP4, EED and

AEBP2 (UniProtKB: Q15910-2, Q15022-1, Q09028-1, O75530-1 and Q6ZN18-2, respectively) into a pFastBac1 expression vectors, modified to include either a PreScission-cleavable N-terminal hexahistidine-MBP tag or TEV-cleavable N-terminal hexahistidine tag, were previously described[36,52,53]. Full-length PALI1 was assembled and subcloned into the pFB1.HMBP.A3.PrS.ybbR vector using Gibson Assembly (primers as indicated in Table S1) with gene synthesised N-terminal fragment (amino acids 1–310) of PALI1 (GeneScript) and commercially available C10orf12 cDNA clone encoding PALI1 amino acids 311–1557 (Millennium Science #MHS6278-202756878) as templates (see Supplementary Table 1 for the full-length PALI1 ORF sequence). The PIR (amino acids 1058-1250) and PIR-long (amino acids 1058-1329) fragments of PALI1 were subcloned into the pFB1. HMBP.A3.PrS.ybbR vector digested with XmaI and XbaI (NEB), under a PreScission-cleavable N-terminal hexahistidine-MBP tag, using Gibson Assembly (NEB #E2611L) with primers as indicated in Supplementary Table 1.

Mutations were introduced to plasmids coding for PALI1 and its truncations using Takara PrimeSTAR HS DNA Polymerase (Clontech #R045A), with primers indicated in Supplementary Table 1. Baculovirus production, titration, infection, and cell harvesting and the purification of PRC2, PRC2-PALI1$_{PIR}$, PRC2-PALI1$_{PIR-long}$ and their mutants performed as previously described[36]. The expression and purification of PRC2 in complexes with MBP-fused PALI1 truncations, PRC2-[MBP-PALI1$_{PIR}$] and PRC2-[MBP-PALI1$_{PIR-long}$] performed as above, with the exception that PRC2 core subunits were expressed under TEV-cleavable hexahistidine-tag and PALI1 truncations under PreScission-cleavable N-terminal hexahistidine-MBP tag. During the purification of these constructs, only TEV was used to cleave tags selectively from the PRC2 core subunits, with the MBP tag on the PALI1 construct remaining intact. All the complexes were snap-frozen in liquid nitrogen and stored at −80 °C as single-use aliquots.

For the structure-function study of EED, two fragments of human EED (amino acids 40–441 and 76–441) were subcloned into a pGEX-MHL expression vector with a TEV-cleavable N-terminal GST-tag (a gift from the lab of Asst. Prof. Yufeng Tong, University of Windsor) using primers as in Supplementary Table 1. The recombinant proteins were overexpressed in *E.coli* BL21 (DE3) at 17 °C overnight and then purified by Glutathione-agarose (Sigma #G4510). Briefly, harvested cells were resuspended in an ice-cold lysis buffer (20 mM Tris-HCl pH 7.5 at 25 °C, 250 mM NaCl, 1 mM phenylmethylsulfonyl fluoride (PMSF) and 1 mM DTT) and lysed using sonication. The cleared lysate was batch-bound to Glutathione-agarose and washed using ice-cold 10 column volumes (c.v.) of lysis buffer before proteins were eluted using ice-cold elution buffer (20 mM Tris-HCl pH 7.5 at 25 °C, 150 mM NaCl, 10 mM reduced glutathione). Tag cleaved using TEV, overnight at 4 °C. The protein was subsequently purified by heparin HP column (GE #17040701), using a buffer containing 20 mM Tris-HCl pH 7.5 at 4 °C and a 150–2000 mM NaCl gradient. Gel filtration purification carried out using HiLoad 16/600 Superdex 200 size exclusion column (GE #28-9893-35), using a buffer containing 20 mM HEPES pH 7.5 and 150 mM NaCl. The peak fractions were pooled, concentrated to a buffer containing 20 mM HEPES pH 7.5, 150 mM NaCl, 1 mM Tris(2-carboxyethyl)phosphine (TCEP) and snapped-frozen as single-use aliquots.

For measuring the DNA binding of PALI1$_{PIR}$, The PIR (amino acids 1058-1250) fragment of PALI1 were subcloned into the pMAL-mhl vector (a gift from the lab of Dr. Yufeng Tong, University of Windsor) digested with BseRI (NEB), under a TEV-cleavable N-terminal MBP tag, using Gibson Assembly (NEB #E2611L) with primers as indicated in Supplementary Table 1. The recombinant proteins were overexpressed in *E.coli* BL21 (DE3) at 17 °C overnight and then purified by Amylose resin (NEB #E8021). The tag was removed and further purified using the same procedure and buffer from EED purification with the exception is that the gel filtration purification was carried out using Superdex 75 Increase 10/300 GL column. The peak fractions were pooled, concentrated to a buffer containing 20 mM HEPES pH 7.5, 150 mM NaCl, 1 mM TCEP and snapped-frozen as single-use aliquots.

For purifying PALI1$_{PIR}$ from the pFB1.HMBP.A3.PrS.ybbR vector, Baculovirus production, titration, infection and cell harvesting was performed as previously described[36]. The purification was performed as previously described[36] with the exception that the N-terminal hexahistidine-MBP tag was retained and the gel filtration purification was carried out using Superdex 75 Increase 10/300 GL column.

**Nucleosome reconstitution.** Recombinant human histones were purified from inclusion bodies and reconstituted into histone octamers as previously described[54] using *E.coli* expression vectors that were a gift of David Tremethick, Australian National University. Cy5-labelled H2A was produced as previously described[55]. Specifically, a single-cysteine mutation (T120C) was introduced into the H2A expression construct. The H2A variant was expressed and purified from inclusion bodies as the wild-type histones. The lyophilised H2A T120C was dissolved in labelling Buffer (20 mM Tris-HCl pH7.0, 6 M guanidine-HCl, 5 mM EDTA) to a final protein concentration of 2 mg/mL and was incubated for 2 h at room temperature. Next, Cy5-maleimide (Lumiprobe #13080) was added to the histone variant solution to a final concentration of 3 mM and the mixture was incubated in the dark for 3 h at room temperature. The labelling reaction was quenched by adding β-mercaptoethanol to a final concentration of 80 mM. Small molecule

reactants were then diluted away by repeat concentrating the solution ≥5 times using 4 mL Amicon Ultra 10,000 MWCO, while toping up the solution by labelling Buffer between concentration cycles.

For the generation of mononucleosomes, 147- or 182-base-pair DNA including one copy of the 601 Widom sequence was PCR amplified. In cases where fluorescently labelled DNA was required for the reconstitution of mononucleosomal construct, for EMSA with a mixture of DNA and a nucleosomal probes (Fig. 6b), 182-base-pair DNA was generated using PCR as described above, except that a 5′-Cy5 linked forwarded primer was used. The template for amplifying 147-base-pair DNA was obtained from Addgene (plasmid #26656) and the template for the amplification of the 182-base-pair DNA was obtained from gene synthesis, with both sequences and the sequences of the primers used for their PCR amplification are indicated in Supplementary table 1. DNA was purified via HiTrap Q HP column (GE #17-1154-01) with 10 c.v. gradient starting with buffer A (20 mM Tris-HCl, pH 7.5 at 25 °C, 150 mM NaCl) into 50% buffer B (20 mM Tris-HCl, pH 7.5 at 25 °C, 2 M NaCl). The peak fractions were pooled, DNA was concentrated by ethanol precipitation, DNA was then dissolved in TE buffer. Mononucleosomes were reconstituted using the salt dialysis method[54]. Reconstituted mononucleosomes were dialysed against a buffer consisting of 20 mM Tris-HCl pH 7.5 (at 25 °C), 25 mM KCl, 1 mM EDTA and 1 mM DTT and concentrated with Amicon Ultra-0.5 mL centrifugal filter (Merck #UFC503096). Nucleosomes were stored at 4 °C and the quality of nucleosomes was assessed by 5–6% TBE gel.

For the generation of nucleosomal arrays, a genomic region (GRCh37/hg19, chr4:94748129-94751759) including the *ATOH1* gene was cloned into pUC18 vector linearised with restriction enzyme SmaI (NEB #R0141). ATOH1 DNA was amplified in a large scale, typically of 10–15 ml, with homemade Pfu DNA polymerase with primers as indicated in Supplementary table 1. The amplified DNA was purified by ion exchange chromatography using Q resin (GE Healthcare). Purified DNA was concentrated by precipitation using iso-propanol and was then dissolved in TE buffer.

Arrays were assembled using the gradient dialysis approach, starting with 0.7 mg/ml ATOH DNA and in the presence of 20-fold molar excess of histone octamers in a dialysis tubing (MWCO: 12–14 kDa, Spectra/Por®4 Dialysis Membrane). The starting buffer was 20 mM Tris-HCl pH7.5, 2 M NaCl, 1 mM EDTA, 1 mM DTT and the buffer was gradually exchanged into 250 mM KCl, 20 mM Tris-HCl pH7.5, 1 mM EDTA, 1 mM DTT over 27 hours at 1 mL/min using peristaltic pump. The final step dialysis was performed into the 20 mM Tris pH7.5, 2.5 mM KCl, 1 mM EDTA, 1 mM DTT. All dialysis were done at 4 °C. The concentration of the nucleosome core particles in the arrays was determined by measuring the concentration of the octamers using BCA assay (ThermoFisher #23252). Arrays were stored at 4 °C and the quality was assessed by 0.8% agarose TBE gel.

**In vitro HMTase activity assays using radiolabelled S-adenosyl-l-methionine.** HMTase activity assays were performed as previously described[36], with some modifications. In brief, each 10 μL HTMase reaction contained 500 nM PRC2, 2 μM mononucleosomes and 5.0 μM S-[methyl-14C]-adenosyl-l-methionine (PerkinElmer, #NEC363050UC) in the presence or absence of stimulatory or control peptides in concentrations as indicated in the text. For HMTase assays in the presence or absence of the PRC2 allosteric inhibitor A395 or the negative control A395N, PRC2 concentration was adjusted to 200 nM and the concentration of other reagents remained the same. For HMTase assays carried out for the identification of PALI1 methylations with the aid of 3C protease, 800 ng protease per reaction was added. All the reactions were incubated for 1 h at 30 °C in buffer containing 50 mM Tris-HCl pH 8.0 at 30 °C, 100 mM KCl, 2.5 mM MgCl₂, 0.1 mM ZnCl₂, 2 mM 2-mercaptoethanol, 0.1 mg/ml BSA (NEB #B9000) and 5% v/v glycerol. The reactions were then stopped by adding 4× LDS sample buffer (Thermo Fisher Scientific, #NP0007) to a final concentration of 1× LDS Samples were then heated at 95 °C for 5 min prior to being subjected to 16.5% SDS-PAGE. Gels were stained with InstantBlue Coomassie protein stain (Expedeon, #ISB1L) before vacuum-drying for 1 h at 80 °C. Dried gels were then exposed to a storage phosphor screen (GE Healthcare) for 1–7 days before acquiring radiograms using Typhoon 5 Imager (GE Healthcare). Densitometry was carried out using ImageJ[56]. All experiments were performed in triplicate.

**Quantitative in vitro HMTase assays with kinetic analysis.** Unless otherwise indicated, 50 nM PRC2 or 15 nM PRC2-PALI$_{PIR}$ were assayed in a 384 reaction plate with varying concentrations of the ATOH arrays, as indicated, in a total reaction volume of 8 μL. The HMTase assay buffer included 50 mM Tris pH 8.0 at 30 °C, 25 μM SAM, 0.5 mM MgCl₂, 0.1% Tween-20, 5 mM DTT and either 35 mM or 100 mM KCl as indicated in the text. Reactions were incubated at 30 °C for 2 h and quenched using 1 μL of 2.4% v/v TFA. For each enzyme and condition, an identical reaction was set without a nucleosomal substrate, in order to account for the automethylation activity of PRC2. Another reaction without the enzyme was carried out to account for a spontaneous conversion of SAM to SAH. To ensure that reaction velocities are measured only under conditions where product formation is linear with time, progress curves were carried out separately, in the presence of the highest substrate concentration applied above during substrate

titration experiments, with the reaction stopped at 10 time points between $t = 0$ to $t = 180$ min. The luminescence signal was developed using reagents supplied with the MTase-Glo™ Methyltransferase Assay (Promega) kit. The plate was read on BMG FLUOstar OPTIMA plate reader (BMG Labtech). $K_M$ and $k_{cat}$ for ATOH arrays were obtained by plotting data into GraphPad Prism, and applying non-linear regression analysis for the Michaelis–Menten model for data fitting. Three independent measurements were performed on three different days.

**Liquid chromatography with tandem mass spectrometry and MS/MS data analysis.** For the identification of methylations in the recombinant PRC2-PALI1$_{PIR-long}$ complex, 0.5 μM protein was incubated in the presence or absence of 20 μM SAM (NEB #B9003) in 25 mM HEPES pH 8.0, 50 mM NaCl, 2 mM MgCl₂, 2 mM 2-mercaptoethanol and 10% glycerol for 1 h at 30 °C, in a total final volume of 85 μL. The proteins were then alkylated, subjected to tryptic digestion and the tryptic peptides where treated and analysed by MS/MS as previously described[36]. Methylated residues were identified using MaxQuant[57] by searching against a database containing the protein subunits of the PRC2-PALI1$_{PIR-long}$ complex.

Publicly available A/IP-MS data (identifiers: PXD004462[27], PXD012547[28], PXD013390[29], PXD012354[26] and PXD003758[15]) were downloaded from ProteomeXchange and used to identify methylations of PRC2 subunits. Methylated residues were identified using MaxQuant[57] by searching against a database containing the human or mouse proteome from Uniprot (Proteome ID UP000005640 or UP000000589, respectively), with the sequences for human or mouse PALI1 and PALI2 appended to them (Uniprot identifiers: Q96JN0-3, A0A1B0GVP4, A0A571BF12 and A0A571BEC4). The amino acid numbering of methylated amino acids throughout the text and figures are based on the human numbering, with the mouse amino acids numbers converted to human numbers for compatibility, based on pairwise sequence alignment[58].

**Crystallisation and structure determination.** Purified EED (amino acids 76-441) protein at 1.5–2.0 mg/mL was mixed with PALI1 peptides at 1:5 EED:PALI1-peptide molar ratio and the mixture were incubated at 4 °C for 1 h before subjected to crystallisation trials. Crystals were grown using the vapour diffusion method in a buffer containing 3.5–3.9 M sodium formate, 10 mM TCEP and 5% glycerol. Crystals were soaked in a reservoir solution with 10–20% glycerol before flash-freezing in liquid nitrogen.

X-ray diffraction data were collected at MX2 beamline of the Australian Synchrotron[59]. All structures were determined by molecular replacement using PHASER[60] within CCP4 package[61], using EED-H3K27me3 structures[23,62] as the search model (PDB: 3JZG[62] and 3IIW[23]). REFMAC5[63] and PHENIX[64] were used for refinement and Coot[65] was used for manual structure building and visualisation. Figures were generated with PyMOL (The PyMOL Molecular Graphics System, Version 2.0 Schrödinger, LLC.). Crystal diffraction data, refinement statistics and PDB accessions for these structures are in Table 1.

**DNA binding assays using fluorescence anisotropy.** 3′-fluorescein-labelled CpG46 and CpG46 mt DNAs were synthesised by Integrated DNA Technologies, Inc. The CpG46 DNA probe was destined to form a hairpin including a 46 bases long double-stranded DNA with a nucleotide sequence originating from a CpG island of the human *CDKN2B* PRC2-target gene, with this sequence mutated in CpG46 mt DNA to remove the CpG sequences and reduce the GC content (for DNA probes sequence see Supplementary Table 1).

DNA was incubated for 2 min at 95 °C in 10 mM Tris-HCl pH 7.5 (at 25 °C) and was immediately snap-cooled on ice for 2 min. Next, DNA was allowed to fold at 37 °C for 30 min in binding buffer (50 mM Tris-HCl pH7.5 at 25 °C, 100 mM KCl, 2 mM 2-mercaptoethanol, 0.1 mg/mL BSA, 0.05% NP-40). In all, 2-fold serial dilutions of protein were made into a binding buffer and were added into the fluorescently labelled DNA probe. The DNA probe concentration was 5 nM for each 40 μL reaction volume and the mixtures were equilibrated at 30 °C for 30 min before measurement. Fluorescence anisotropy data were collected using a PHERAstar plate reader (BMG Labtech) at 30 °C. The background was subtracted from protein-free samples. Data were fitted with GraphPad Prism 8 software using non-linear regression for specific binding with a Hill slope function. All experiments were performed in triplicates that were carried out on different days.

**RNA-binding assays using fluorescence anisotropy.** The RNA-binding affinities of proteins were quantified using fluorescence anisotropy. Experiments were carried out as we previously described[36]. Briefly, a 3′ fluorescein-labelled G4 24 RNA with the sequence (UUAGGG)₄ was used and the reaction took place as above, with the exception that the initial incubation at 95 °C was limited to 1 min and the binding buffer was 50 mM Tris-HCl pH7.5 at 25 °C, 200 mM KCl, 2.5 mM MgCl₂, 0.1 mM ZnCl₂, 2 mM 2-mercaptoethanol, 0.1 mg/mL BSA, 0.05% NP-40 and 0.1 mg/mL fragmented yeast tRNA (Sigma, #R5636).

**Peptide binding assays using fluorescence anisotropy.** For assaying the affinity of the JARID2-K116me3 peptide for EED, various concentrations of EED (amino acids 40–441) were incubated with 40 nM of 5-FAM labelled JARID2-K116me3

peptide in binding buffer (50 mM Tris-HCl pH 7.5 at 25 °C, 100 mM KCl, 2 mM 2-mercaptoethanol, 0.1 mg/mL BSA, 0.05% NP-40, 2.5% glycerol) at 30 °C for 30 min before a fluorescence anisotropy measurement took place using a PHERAstar plate reader. Data processing was carried out as previously described[66], with some modifications. Specifically, with changing the concentration of EED protein (P), we recorded △Robs: the observable anisotropy of the mixtures after the subtraction of the observable anisotropy of the 5-FAM labelled JARID2-K116me3 peptide ligand. Data were fitted to equation (1), below, by using non-linear least-squares fit (Matlab, MathWorks) to estimate the anisotropy difference △r and the dissociation constant between the protein EED to the ligand JARID2-K116me3 peptide, $K_L$:

$$\triangle Robs = \frac{\triangle r[PL]}{[PL]+[L]} \qquad (1)$$

where the concentrations of the protein-ligand complex $[PL]$ and the free ligand $[L]$ are calculated from Eqs. (2) and (3) below, respectively, with $P_0$ and $L_0$ indicates the total concentration of the protein and the ligand, respectively.

$$[PL] = \frac{(K_L + [P_0] + [L_0]) - \sqrt{(K_L + [P_0] + [L_0])^2 - 4[P_0][L_0]}}{2} \qquad (2)$$

$$[L] = [L_0] - [PL] \qquad (3)$$

Fluorescence anisotropy displacement titrations were used to assay the dissociation constants of the unlabelled peptides (N) and the protein (P). Assays were carried out as described above, with the exception that 2-fold serial dilutions of unlabelled peptides were combined with EED at a final concentration of 10 μM and 5-FAM labelled JARID2-116me3 peptide at a final concentration of 40 nM before anisotropy data were collected as described above. Data processing was carried out as previously described[66]. Specifically, △Robs were recorded for each peptide [N] concentration point and Eq. (1) used to estimate $K_N$ and △r as the fitting parameters. $K_N$ is the equilibrium dissociation constant for binding of the unlabelled peptide $N$ to the protein $P$. $N_0$ is the total concentration of $N$. $[PL]$ and $[L]$ are calculated in a different way:

$$[PL] = \frac{[L_0][P]}{K_L + [P]} \qquad (4)$$

and $[L] = [L_0] - [PL]$

where $K_L$ are obtained from the measurement above, and $[P]$ is one of the roots of the cubic equation:

$$[P]^3 + c_2[P]^2 + c_1[P] + c_0 = 0 \qquad (5)$$

where

$$c_2 = K_L + K_N + [N_0] + [L_0] - [P_0] \qquad (6)$$

$$c_1 = K_L K_N - K_N[P_0] + K_L[N_0] - K_L[P_0] \qquad (7)$$

$$c_0 = -K_L K_N[P_0] \qquad (8)$$

**Electrophoretic mobility shift assay.** Cy5-DNA or Cy5-H2A labelled nucleosomes were diluted using binding buffer (50 mM Tris-HCl, pH7.5 at 25 °C, 100 mM KCl, 2 mM 2-mercaptoethanol, 0.05% v/v NP-40, 0.1 mg/mL BSA, 5% glycerol, with additional 5 ng/μL Salmon Sperm DNA for measuring binding for Cy5-H2A labelled nucleosomes). In total, 2-fold serial dilutions of protein in binding buffer were combined with nucleosome probes, to a final probe concentration of 5 nM. The reaction mixtures were incubated at 4 °C for 30 min and then subjected to non-denaturing gel electrophoresis at 6.6 V/cm over a 0.7% agarose gel buffered with 1× TBE at 4 °C for 1.5 h. Gels were imaged using Typhoon 5 Imager (GE Healthcare) to record signals from the Cy5 dye. The fractions of bound Cy5-DNA labelled nucleosomes were calculated based on the unbound nucleosomes band and the fractions of bound Cy5-H2A labelled nucleosomes were calculated based on the bound nucleosomes band, with the densitometry analysis carried out using ImageJ[56]. Data were fitted with GraphPad Prism software using non-linear regression for specific binding with Hill slope function. All experiments were performed in triplicate.

For measuring the affinity of PALI1_PIR to DNA, the DNA probe was incubated for 2 min at 95 °C in 10 mM Tris-HCl pH 7.5 (at 25 °C) and was immediately snap-cooled on ice for 2 min. Next, DNA was allowed to fold at 37 °C for 30 min in binding buffer (50 mM Tris-HCl pH 7.5 at 25 °C, 100 mM KCl, 2 mM 2-mercaptoethanol, 0.1 mg/mL BSA, 0.05% NP-40 and 5% glycerol). In all, 2-fold serial dilutions of PALI1_PIR were made into a binding buffer and were added into the fluorescently labelled DNA probe to a final DNA probe concentration of 5 nM. The mixtures were equilibrated at 30 °C for 30 min and then subjected to non-denaturing gel electrophoresis at 6.6 V/cm over a 0.7% agarose gel buffered with 1× TBE at 4 °C for 1.5 h. Gels were imaged using Typhoon 5 Imager (GE Healthcare) to record signals from the fluorescein dye. The fractions of DNA bound were calculated based on the unbound DNA bands, with the densitometry analysis carried out using ImageJ[56]. Data were fitted with GraphPad Prism software using non-linear regression for specific binding with Hill slope function. All experiments were performed in triplicate.

**Cell culture.** K562 cells were cultured in RPMI-1640 (Merck #R8758) growth medium and HEK293T and HeLa were cultured in DMEM growth medium. In all cases, growth media were supplemented with 10% FBS (Cellsera AU-FBS/SF) and 1% (v/v) penicillin-streptomycin (Thermo Scientific #15140122) and incubated at 37 °C with 5% CO2. K562 and HeLa cells were acquired from ATCC and cells were tested periodically for mycoplasma contamination.

**Plasmid transfection, generation of lentiviruses and lentiviral transduction.** Flag-PALI1 WT and K1241A mutant, and Flag-LacZ (ORF originated from Addgene #25893), were subcloned into SmaI (NEB #R0141) linearised pHIV-EGFP (Addgene #21373) or pHIV-dTomato (Addgene #21374) vectors using Gibson Assembly (see Table S1 for primers) and NEB stable Competent E.coli (NEB #C3040). For plasmid transfection, ~10^6 HEK293T or HeLa cells were seeded in a six-well plate. The following day, the medium was replaced with 2 ml of antibiotic-free DMEM. The transfection mixture for HEK293T cells was created by adding 9 μL Lipofectamine™ LTX Reagent with 3 μL PLUS™ Reagent (Thermo Scientific #15338100) and 3 μg of a plasmid to 500 μL Opti-MEM™ (Thermo Scientific #31985062). The transfection mixture for HeLa cells instead consisted of 6 μL Lipofectamine™ LTX Reagent with 3 μL PLUS™ Reagent and 1.25 μg of transfer plasmid in 500 μL Opti-MEM™ (Thermo Scientific #31985062). The transfection mixture was incubated at room temperature for 25 min and then added to the cells before returning them to the incubator. The growth medium was replaced after 24 h. For cell harvesting ahead of immunoblotting, 48 h after transfection the media was removed and replaced with 100–400 μL of Laemmli buffer (1% (v/v) SDS, 12.5% (v/v) glycerol, 35 mM Tris pH 7.5 at 25 °C, 0.01% (w/v) bromophenol blue, 5 mM MgCl₂, 1% (v/v) 2-mercaptoethanol) and 25 U/mL Benzonase (Merck #70746).

For the generation of lentiviruses, HEK293T cells were transfected as above, with 0.5 μg pMD2.G plasmid (Addgene #12259), 1 μg psPAX2 plasmid (Addgene #12260) and 1.5 μg of transfer plasmid. After 48 and 72 h, the culture supernatant containing the lentivirus was collected and stored at −80 °C. For lentiviral transduction, 250 μL of lentiviral supernatant was added to 3 × 10^4 K562 cells to a final volume of 550 μL, with polybrene at a final concentration of 8 μg/mL.

**Immunoblotting.** For nuclear fractionation of K562 cells, cells were washed twice with phosphate-buffered saline (PBS) by centrifugation at 500 × g for 5 min then resuspended in cytoplasmic extraction buffer (20 mM Tris pH 7.5 at 25 °C, 0.1 mM EDTA, 2 mM MgCl₂, 20 mM BME and protease inhibitor cocktail (Sigma #4693132001)) to a density of 2 × 10^7 cells/mL. The cells were incubated for 2 min at room temperature then 10 min on ice before adding NP-40 to a concentration of 1% (v/v) and mixing. Samples were centrifuged at 4 °C and 500 × g for 3 min and the supernatant was kept as the cytoplasmic fraction. The nuclear fraction was washed in cytoplasmic extraction buffer with 1% (v/v) NP-40 twice, by centrifugation at 4 °C at 500 × g for 3 min, and was then resuspended in Laemmli buffer.

Samples containing 50 μg total protein were loaded on a 10%, 16.5% or 8-16% acrylamide gel (Biorad #4561103) for SDS-PAGE and then transferred to a nitrocellulose membrane (GE Life Sciences #10600002). Membranes were incubated in blocking buffer (Thermo Scientific #37539) for 1 hour at room temperature before applying antibodies. Signal was developed using SuperSignal™ West Pico PLUS Chemiluminescent Substrate (Thermo Scientific #34580) and images were taken on a ChemiDoc™ imager. All experiments were performed in triplicates.

The antibodies used for immunoblotting include: anti-Flag HRP-conjugated (Sigma #A8592, 1:1000), anti-LCOR (Merck #ABE1367, 1:250), anti-actin (Sigma #A2066, 1:800), anti-EZH2 (Active Motif #39875, 1:10000), anti-EED (Abclonal #A12773, 1:5000), anti-H3 (Abcam #Ab1791, 1:100000), anti-H3K27me3 (Merck #07-449, 1:5000 or Cell Signalling #9733 S, 1:4000), anti-mouse HRP-conjugated (Jackson Immuno-Reasearch #715-035-150, 1:5000) and anti-rabbit HRP-conjugated (Santa Cruz Biotechnology #sc-2357, 1:5000).

**Competitive cell proliferation assay.** Cells were transduced with lentiviruses carrying the specified gene constructs and were then cultured, each sample separately, for 7 days. At this point, an equal number of eGFP or dTomato positive cells were sorted using flow cytometry, as described below, and combined into the same collection tube and placed in the same well for the competition experiment. Cells of the two competing treatments were cultured together in the same well for 7 days. Next, the number of eGFP- and dTomato-expressing cells counted using flow cytometry with the B530-A and YG586-A detectors, respectively. Three independent biological replicates, starting from lentivirus transduction, were initiated on three different days and were carried out as described above.

**Detection of eGFP and dTomato using Flow cytometry.** Before sorting or analysis by flow cytometry, cells were centrifuged at 500 × g for 5 min, and the supernatant removed. The cells were then resuspended in flow cytometry buffer (PBS supplemented with 10% FBS and 615 μM EDTA) to a density of ~10^7 cells/mL, ran through a cell strainer (Falcon #352235) and kept on ice. For the detection of GFP or dTomato, the cells were sorted by flow cytometry on a BD Influx™ cell sorter using the 488 nm or the 561 nm lasers, respectively. For the selection of

transduced cells, and unless stated otherwise, gates for sorting were set to include the top 10 or 20% of the GFP or dTomato positive cells in the samples transduced with PALI1 wild-type, based on the first replicate. For the analysis of transduced cells, ~0.5–1.0 × $10^5$ intact single cells were analysed on a BD LSRFortressa™ X-20 analyser, where the threshold for GFP or dTomato positive cells was defined based on the intensity observed at the top 0.1% of untransduced K562 cells. Gating strategies used for cell sorting are shown in Supplementary Fig. 11. Data were analysed using BD FACSDiva™ and GraphPad Prism.

**Detection of CD235a or CD44 using Flow Cytometry**. Cells were transduced with lentiviruses carrying the specified gene constructs and were then cultured for 7 days. Cells were sorted for high GFP or dTomato expression, as described above, and then cultured for 7 additional days before the growth medium was removed by spinning the cells at 500 × g for 5 min. Cells at a density of 2 × $10^7$ cells/mL were incubated on ice for 15 min in flow cytometry buffer with 2.5 μL of Pacific Blue™ conjugated anti-CD235a antibody (BioLegend #349108) per 100 μL for detection of CD235a, or 2.5 μL of APC or PE conjugated anti-CD44 antibody (Biolegend #103011, Biolegend #103007) per 100 μL for the detection of CD44. The cells were centrifuged again at 500 g for 5 minutes and the supernatant removed, then washed with antibody-free flow cytometry buffer. The cells were then analysed by flow cytometry for the quantification of CD235a or CD44 in the GFP or dTomato positive cells using the V450-A and R670-A or YG586-A detectors, respectively. Three independent biological replicates, starting from lentivirus transduction, were initiated on three different days and were carried out as described above. The data was analysed using FlowJo and GraphPad Prism.

**Doxycycline-induced CRISPR/Cas9 knockout of EED**. K562 cells were first transduced with lentiviruses generated using a vector for expression of Cas9 with blasticidin selection (Addgene #52962), and then selected using 10 μg/mL of blasticidin, and were continuously grown in the presence of that antibiotic. For the doxycycline-induced expression of the gRNA, DNA oligos with the sequence of the target site in EED (Supplementary Table 1) were annealed together and cloned as previously described[67] into the FgH1tUTG vector (Addgene #70183, was a gift from Marco Herold) to form the FgH1tUTG.EED plasmid. The FgH1tUTG.EED plasmid was packed into lentiviruses, as described above, which were then used to transduce K562 cells that were next selected using flow cytometry, based on EGFP expression. Expression of the gRNA was induced using 3 μg/mL of doxycycline (Merck #631311).

**Chromatin immunoprecipitation**. HEK293T cells were transfected in one 10 cm dish per two IP samples, with the transfection carried out by scaling the transfection process described above (in "Plasmid transfection, generation of lentiviruses and lentiviral transduction"). The next day, the cells were passed into 15 cm dishes. In all, 48 h after transfection, the cells were washed with room temperature PBS, and then crosslinked with 1% (v/v) formaldehyde in PBS for 10 min at room temperature. The formaldehyde was quenched by adding glycine to a final concentration of 0.125 M for 2 min. After removing the buffer, cells were scraped from the plate and transferred into 50 mL of ice-cold PBS, then centrifuged at 1000 RCF for 5 min and washed once using PBS. After the PBS was removed, the cell pellets were stored at −80 °C.

The cell pellets were lysed in 800 μL lysis buffer (1% (v/v) SDS, 10 mM EDTA, 50 mM Tris-HCl pH 8 and 1X protease inhibitor cocktail (Sigma #4693132001)) on ice. Chromatin was sheared by sonication at 4 °C in a Bioruptor set to high, with 20 cycles of 30 s on and 30 s off. The samples were then centrifuged at max speed for 10 min at 4 °C. For each antibody, 400 μL of lysate was added to 1.6 mL of dilution buffer (165 mM NaCl, 0.01% (v/v) SDS, 1.1% (v/v) Triton X-100, 1.2 mM EDTA, 16.7 mM Tris-HCl pH 8 and 1X protease inhibitor cocktail (Sigma #4693132001)), and precleared with 50 μL Dynabeads Protein G (Thermo #10003D) for 1 h at 4 °C, rotating at low speed. The beads were removed and the supernatants were incubated with the appropriate antibody overnight at 4 °C, rotating at low speed.

Each sample was washed twice in dilution buffer, low salt buffer (150 mM NaCl, 0.5% (v/v) sodium deoxycholate, 0.1% (v/v) SDS, 1% (v/v) NP-40, 1 mM EDTA, 50 mM Tris-HCl pH 8 and 1X protease inhibitor cocktail (Sigma #4693132001)), and high salt buffer (500 mM NaCl, 0.5% (v/v) sodium deoxycholate, 0.1% (v/v) SDS, 1% (v/v) NP-40, 1 mM EDTA and 50 mM Tris-HCl pH 8), with 10 min incubation at 4 °C, rotating at low speed between each wash. The beads were then washed once in 1 mL of TE buffer (0.25 mM EDTA, 10 mM Tris-HCl pH 8) and transferred into new tubes, with 5 min incubation. To elute, the beads were twice resuspended in 50 μL of elution buffer (1% (v/v) SDS, 100 mM NaHCO3) and vortexed for 15 min at room temperature, and the two eluates were combined. In all, 6 μL of 5 M NaCl and 5 μL of RNase A (Thermo #EN0531) were added to each sample and incubation was carried out at 65 °C with shaking overnight to reverse the crosslinking. In total, 5 μL of Proteinase K (NEB #P8107S) was added to each sample, and they were then incubated for 1–2 h at 60 °C with shaking. DNA was purified from these samples using the Qiagen MinElute PCR purification kit (Qiagen #28006) and eluted in 50 μL. The only change from the manufacturer's instructions was that the binding and elution steps were carried out twice per sample.

DNA was quantified using qPCR in a 10 μL reaction volume in a Biorad CFX384™ Real-Time C1000 Touch™ Thermal Cycler. Each reaction contained 2.5 μL of template DNA that was diluted 1:10, 1.25 μL of 2.8 μM forward primer, 1.25 μL of 2.8 μM reverse primer (0.7 μM final concentration per primer), 5 μL QuantiNova SYBR Green 2X master mix (Qiagen #208054) and was prepared using Qiagen QIAgility. For each biological replicate, Cq values from two technical replicates were averaged from both the input and IP samples, then the following formula was used to calculate the fraction of DNA in the IP: 2^(-(Cq[IP]-Cq [input]) × ((input volume)/(IP volume)). Three independent biological replicates, starting from transfection, were initiated on three different days and were carried out as described above. The data was analysed using FlowJo and GraphPad Prism. The antibodies used for ChIP include: anti-LCOR (Merck #ABE1367, 3 μg), anti-EZH2 (Active Motif #39875, 6 μg), anti-H3K27me3 (Cell Signalling #9733 S, 10 μL) and anti-Flag (Merck #F1804, 3 μg).

**Reporting summary**. Further information on research design is available in the Nature Research Reporting Summary linked to this article.

## Data availability
Coordinates and structure factors have been deposited in the Protein Data Bank under accession codes 6V3X and 6V3Y. The mass spectrometry data have been deposited to Monash University research repository Figshare, with https://doi.org/10.26180/14752509. Source data are provided with this paper.

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

## Acknowledgements

We would like to thank the Monash Biomedical Proteomics Facility for providing instrumentation and technical support, Monash FlowCore for sorting of cells and for assistance with flow cytometry analysis and Monash Molecular Crystallization Facility for initial crystallisation screening. This research was undertaken in part using the MX2 beamline at the Australian Synchrotron, part of Australia's Nuclear Science and Technology Organisation (ANSTO), and made use of the Australian Cancer Research Foundation (ACRF) detector. We also thank the support from the MASSIVE HPC facility (www.massive.org.au). We thank Ruby Law and her lab members for advice, discussions, training and assistance with the X-ray crystallography data collection and structure determination. Q.Z. holds an Australian Research Council (ARC) Discovery Early Career Researcher Award (DE180100219). S.C.A. is supported through an Australian Government Research Training Program (RTP) Scholarship. B.M.O. is supported through an Australian Government RTP Scholarship and also by the Monash Graduate Excellence Scholarship. C.D. is an EMBL-Australia Group Leader and a Sylvia and Charles Viertel Senior Medical Research Fellow and acknowledges support from the ARC (DP190103407) and the NHMRC (APP1162921 & APP1184637).

## Author contributions

Q.Z., S.C.A., S.F.F. and M.U. prepared reagents. Q.Z., S.C.A., M.U., V.L. and B.M.O. developed assays. Q.Z., S.C.A., S.F.F. and V.L. carried out experiments. Q.Z., S.C.A. and S.F.F. analysed data. Q.Z., S.C.A. and C.D. wrote the manuscript. Q.Z., S.C.A. and C.D. designed the project. C.D. supervised the project.

## Competing interests

The authors declare no competing interests.
