## [Peer Review File · Nature Communications]

REVIEWER COMMENTS

Reviewer #1 (Remarks to the Author):

Comments to Zhang et al.

In the manuscript "Convergent evolution between PALI1 and JARID2 for the allosteric activation of PRC2" The authors describe the methylation of the PRC2.1 specific accessory subunit PALI1 and show that the methylated PALI1 protein is able to allosterically activate the KMT activity of PRC2. In vitro competition assays as well as X-ray crystallography suggest that methylated PALI1 and JARID2 compete for the same binding site for a methylated lysine on the EED subunit of the PRC2 complex. In addition to the allosteric activation, PALI1 also increases the affinity of the PRC2 complex to its nucleosomal substrate, most likely by its ability to bind DNA. Finally, the authors show that an overexpression of PALI1 reduces proliferation and stimulates differentiation of K562 cells along the erythroid lineage, which is at least partly dependent on PALI1s ability to allosterically activate the PRC2 complex.

The paper addresses an interesting regulatory aspect of the PRC2 complex and suggests multiple ways of de novo H3K27me3 methylation. I think the presented findings are of interest to the readers of Nature Communication.

The paper is well written (with a few odd spelling mistakes i.e. towered instead of toward) and most of the conclusions are well founded by the data provided. There are a couple of aspects, I would have liked the authors to address in addition to what they show (i.e. does the overexpression of PALI1 result in a faster spreading at certain sites in the genome? Is this dependent on the K1241?) but that may be a bit too much.

I do have, however, a few points that would improve the manuscript and which should not be too difficult for the authors to address.

DETAILED CRITIZSM

Figure 1: Is the autoradiogram shown in figure 1C from the gel shown below? The autorad lacks MW markers and the Coomassie stain lacks a labelling of the bands. The additional band in lane 3 is not labelled but does not seem to be PALI1_pir based on the autorad. Is this the 3C protease?

Figure 2: It would be nice to also show the upper part of the gel in the original figure and not only in the supplement. I would expect to see the wt PALI1 methylated but not the AAA mutation. However, this is hard to see in the supplemental figure as methylation is very weak and the bands are not labelled. It also looks as if only the methylation of nucleosomes is stimulated but not the methylation of Ezh2 and the other bands detected. It would be nice if the authors could comment on this. Finally, the Coomassie stain would serve as a control showing that equal amounts of the PRC2 complex are present. To better understand the mechanism of allosteric activation it would also be nice to see an extended titration experiment with the peptide. Is it just required to kick start and the rest of the activation can be taken over by H3K27me3?

Figure 4: The titration experiment lacks a control with PALI1 only. Does PALI1 bind DNA or nucleosomes on its own or does it just increase the affinity of PRC2 for nucleosomes?

Figure 5: Is the effect seen by overexpression of PALI1 dependent on the presence of PRC2? What happens if an inhibitor of EZH2 is used during overexpression of PALI1?

Reviewer #2 (Remarks to the Author):

In the current study Zhang et al. reported Pali1 methylation by PRC2 and identified sites of methylation on Pali1. The authors also found methylated Pali1 mediated allosteric activation of PRC2 by binding to the Eed subunit directly. The mechanism of allosteric activation was further studied by in vitro methylation assays and structural studies. Finally, the authors tried to connect Pali1 methylation and methyl Pali1-mediated allosteric activation of PRC2 to K562 leukemia cell differentiation. Although the major finding here that methylated Pali1 activates PRC2 activity is new, the overall novelty of this study is limited at least in the current format, in light of what is already known in the literature.

Pali1-mediated PRC2 activation was previously shown by Conway et al. (Mol Cell. 2018). In the same study, loss of Pali1 was reported to reduce H3K27me3 enrichment on PRC2 targets. Data from the current study further indicated that Pali1 could contribute to PRC2 activation via two distinct mechanisms, including enhanced nucleosome binding as well as allosteric activation through Eed. The latter mechanism required direct interaction of methyl Pali1 with Eed. The structural study on Eed in complex with methylated Pali1 peptides revealed structures very similar to that of Eed bound to a methylated Jarid2 peptide, which was reported a few years ago by Sanulli et al. (Mol Cell. 2015). In that earlier paper, methylated Jarid2 was also indicated to regulate cell differentiation based on a series of in vitro methylation assays and ChIP analysis. Pali1 was not the first non-histone substrate ever identified. In addition to Jarid2 mentioned above, Ezh2 and Elongin A were both recently found to be methylated by PRC2 to regulate H3K27 methylation or transcription.

Although the authors observed that overexpression of the Pali1-K1241A mutant in K562 cells failed to promote differentiation as compared with wildtype Pali1, the mechanism was completely lacking. Was the H3K27me3 level changed on chromatin loci implicated in the regulation of differentiation? Was such change if any caused by the lack of allosteric activation of PRC2?

Publication is not recommended unless major revision can be completed to add novelty and to provide mechanistic insights on the functional study.

Major points:

Throughout the manuscript, the authors used the truncated PRC2-Pali1PIR complex for the in vitro methylation and DNA/nucleosome binding assays. Some of the major conclusions need to be confirmed with full-length PRC2-Pali1 complex.

In Fig. 4b, it is unclear why the authors chose to use a mixture of tailed mononucleosomes and free DNA for EMSA. In addition, what is the binding affinity between PRC2-Pali1 and the nucleosome core without the tail? It is difficult to quantitate the contribution of the nucleosome core to the binding when the DNA tail is present.

Is there experimental evidence to show that Pali1 methylation occurs prior to H3K27 methylation in cells? Otherwise, what would be the functional importance for methyl Pali1-mediated PRC2 activation when existing H3K27me3 can also activate PRC2?

The authors used an overexpression system to compare the effect of wildtype Pali1 and the Pali1-K1241A mutant on K562 cell differentiation. The authors showed roughly comparable levels of Pali1 expression by Western blot. However, it is unclear how sensitive the observed cell phenotype is to the protein expression level. If even a subtle difference in protein expression may cause a dramatic change of the phenotype, the current experiment needs to be better calibrated.

The present cell proliferation and differentiation marker expression data are insufficient to conclude that methyl Pali1-mediated PRC2 activation has an important function in cell differentiation. Given that the global H3K27me3 level stayed unchanged for cells expressing Pali1-K1241A, it is necessary to investigate by ChIP analysis whether the H3K27me3 level on specific PRC2 targets is affected by this mutant. Chromatin targeting of PRC2-Pali1 needs to be studied in parallel as a control.

Reviewer #3 (Remarks to the Author):

Polycomb repressive complex 2 (PRC2) is a histone methyltransferase that is essential for maintaining cellular identity during development and can be divided into two major classes, PRC2.1 and PRC2.2. PRC2.2 contains the accessory subunit, JARID2 which is known to trigger an allosteric activation of histone methyltransferase. However, it is not known which specific subunits of PRC2.1 is the counterpart of JARID2 to facilitate PRC2 activity. This manuscript explores the roles and molecular mechanisms of the PRC2.1 specific subunit, PALI1 in an allosteric activation of PRC2.1. Authors indicate that the PRC2-binding domain of PALI1 enhances H3K27 methyltransferase activity of PRC2.1 by two independent mechanisms. (1) DNA- and substrate binding and (2) allosteric stimulation. The mechanistic resemblance between PALI1 and JARID2 indicates both proteins regulate allosterically activation of PRC2.1 and PRC2.2 complex respectively. Although most of experiments are done carefully and the conclusions are well supported by the results, in vivo relevance of the findings is limited. I do have the following comments which I hope will improve the impact of the study:

Major points

1) It is unclear to me regarding complexity of PRC2.1. PRC2.1 contains one of PCL proteins together with either EPOP or PALI1. If EPOP and PALI1 form a complex with PRC2 in a mutually manner, how does EPOP-containing PRC2 achieve the stimulation of PRC2? In line with this aspect, does PALI2 have a similar property to PALI1 in terms of conserved Lysine residues that potentially undergo methylation and stimulation of DNA binding and activity of PRC2? Also, If PALI1 facilitates chromatin binding of PRC2, what is the function of PCL proteins? Authors should discuss in the manuscript.

2) Authors convincingly demonstrated that PALI1 K1241 is methylated in vitro and in vivo, and that this methylation facilitates DNA binding of PRC2 and the methyltransferase activity toward H3K27. However, it is unclear about (1) the timing and (2) the spatial constraint of methylation of PALI1 in vivo. If the antibody against PALI1 K1241me_{2/3} or at least the antibody against methyl-lysines are available, authors should perform a biochemical purification of PRC2 from the fractionated cell lysates (cytosol, Nuclear, and chromatin bound) and perform western blotting to assess which fractions are enriched with methylated PALI1.

3) If the authors propose a functional resemblance between PALI1 to JARID2, I would suggest to assess the importance of the +1 adjacent aromatic residue (PALI1 F1242A) in both EED binding and the stimulation of PRC2.

4) In Figure 5C and 5d, authors should perform H3K27me₃ and PRC2 ChIP-qPCR at couple of the PRC2 target loci in the erythroid lineage to investigate that de novo targeting of PRC2 and the H3K27me₃ deposition to a target gene is facilitated by PALI1 WT but not PALI1 K1241A.

Minor comments

1) I think the title of this manuscript is a little bit overstated. While the authors revealed the functional similarities between LARID2 and PALI1, they did not show the evidence of convergent evolution between them.

2) Related to Fig.1, the authors should show if PALI1 is methylated by PRC2 in vivo.

3) In Figure 4b, authors argued that PALI1PIR facilitates DNA binding by PRC2. While, in the manuscript at line 263-266, it is described that the probe can detect interaction at either the centered or the off-centered position, or the free DNA, the conclusion from this experiment is that the PRC2PIP increased the affinity of PRC2 to nucleosome. How about the binding of the free DNA and either the center or off-center?

4) Related to line 286, the authors should discuss the current knowledge of the expression pattern and the prognostic role of PALI1 in myeloid malignancies.

5) In Figure 5 where the authors discussed the biological significance of enhancement of PRC2 catalytic activities by PALI1, the experiments related to PALI1 knock out should be included.

We thank the reviewers for dedicating their time to review the manuscript. Our response to each of the point is specified below.

Reviewer #1 (Remarks to the Author):

Comments to Zhang et al.

In the manuscript “Convergent evolution between PALI1 and JARID2 for the allosteric activation of PRC2” The authors describe the methylation of the PRC2.1 specific accessory subunit PALI1 and show that the methylated PALI1 protein is able to allosterically activate the KMT activity of PRC2. In vitro competition assays as well as X-ray crystallography suggest that methylated PALI1 and JARID2 compete for the same binding site for a methylated lysine on the EED subunit of the PRC2 complex. In addition to the allosteric activation, PALI1 also increases the affinity of the PRC2 complex to its nucleosomal substrate, most likely by its ability to bind DNA. Finally, the authors show that an overexpression of PALI1 reduces proliferation and stimulates differentiation of K562 cells along the erythroid lineage, which is at least partly dependent on PALI1s ability to allosterically activate the PRC2 complex.

The paper addresses an interesting regulatory aspect of the PRC2 complex and suggests multiple ways of de novo H3K27me3 methylation. I think the presented findings are of interest to the readers of Nature Communication.

The paper is well written (with a few odd spelling mistakes i.e. towered instead of toward) and most of the conclusions are well founded by the data provided. There are a couple of aspects, I would have liked the authors to address in addition to what they show (i.e. does the overexpression of PALI1 result in a faster spreading at certain sites in the genome? Is this dependent on the K1241?) but that may be a bit too much.

I do have, however, a few points that would improve the manuscript and which should not be too difficult for the authors to address.

DETAILED CRITIZSM

Figure 1: Is the autoradiogram shown in figure 1C from the gel shown below? The autorad lacks MW markers and the Coomassie stain lacks a labelling of the bands. The additional band in lane 3 is not labelled but does not seem to be PALI1_{pir} based on the autorad. Is this the 3C protease?

Yes, the radiogram recorded from the same Coomassie-stained gel. We now labelled all the bands in both radiogram and Coomassie-stained gel, including the 3C protease, and marked the molecular weight in both of them (Fig 1c).

Figure 2: It would be nice to also show the upper part of the gel in the original figure and not only in the supplement. I would expect to see the wt PALI1 methylated but not the AAA mutation. However, this is hard to see in the supplemental figure as methylation is very weak and the bands are not labelled.

This is now done: we now have replaced the figure 2a with the HMTase assay performed using the complex PRC2-PALI1_{PIR}, which is more stable than the PRC2-PALI1_{PIR-long} that was shown in the original figure. The figure now shows the methylation of PALI1_{PIR}. The gel from the original figure, using PRC2-PALI1_{PIR-long}, has been moved to supplementary Fig. 2c.

It also looks as if only the methylation of nucleosomes is stimulated but not the methylation of Ezh2 and the other bands detected. It would be nice if the authors could comment on this.

We now added a couple of sentences to discuss this point in the Discussion section, under "*PALI1 provides PRC2 with means to gauge its own enzymatic activity before adding a stimulus*" (L416-418 in the revised main text).

Finally, the Coomassie stain would serve as a control showing that equal amounts of the PRC2 complex are present.

We now included in Fig 2a two coomassie stained gels: one demonstrates the purity of all the PRC2-PALI1 mutant complexes and another one is the (dry) gel used to generate the radiogram.

To better understand the mechanism of allosteric activation it would also be nice to see an extended titration experiment with the peptide. Is it just required to kick start and the rest of the activation can be taken over by H3K27me3?

We have now included a titration experiment using PALI1-K1241me3 peptide (Fig. 3b, c). The result nicely complements the peptide-binding assays in the same figure (Fig 3a), with the relative HMTase activity reaching saturation slightly after the *K_d* concentration of the peptide (marked by a vertical dashed line in Fig 3c). The linear dependence between peptide concentration to the relative HMTase activity implies that the PALI1 peptide is insufficient to kickstart the histone methyltransferase

activity of PRC2 toward mono-nucleosome substrates *in vitro*. This could be because the entire PRC2-binding domain from PALI1 is required for high-affinity interactions with nucleosomes (Fig 7b, c).

Figure 4: The titration experiment lacks a control with PALI1 only. Does PALI1 bind DNA or nucleosomes on its own or does it just increase the affinity of PRC2 for nucleosomes?

We have now included an EMSA for measuring the DNA binding activity of PALI1_{PIR} alone (Supplementary Fig. 6c). We found that PALI1_{PIR} itself can bind DNA, even though the affinity is not as high as that of the PRC2-PALI1_{PIR}. We now also show that a PALI1_{PIR} protein that was purified independently of PRC2 is sufficient to enhance the HMTase activity of the core PRC2 complex (Supplementary Fig 2h).

Figure 5: Is the effect seen by overexpression of PALI1 dependent on the presence of PRC2? What happens if an inhibitor of EZH2 is used during overexpression of PALI1?

We now repeated the PALI1 overexpression with FACS sorting in the presence and absence of a doxycycline-inducible knockout of the PRC2 core subunit EED (Supplementary Fig. 9d, e, f). Our findings indicate that both changes in the expression of CD markers (Supplementary Fig. 9d, e) AND also the stability of the ectopically expressed PALI1 (Supplementary Fig. 9g) are dependent on PRC2. Importantly, the loss of ectopically expressed PALI1 upon EED knockout indirectly implies that the vast majority of the ectopically expressed PALI1 molecules in these cells are bound to PRC2. These results are now discussed in L374-376 and L437-439 of the revised manuscript.

Reviewer #2 (Remarks to the Author):

In the current study Zhang et al. reported Pali1 methylation by PRC2 and identified sites of methylation on Pali1. The authors also found methylated Pali1 mediated allosteric activation of PRC2 by binding to the Eed subunit directly. The mechanism of allosteric activation was further studied by *in vitro* methylation assays and structural studies. Finally, the authors tried to connect Pali1 methylation and methyl Pali1-mediated allosteric activation of PRC2 to K562 leukemia cell differentiation. Although the major finding here that methylated Pali1 activates PRC2 activity is new, the overall novelty of this study is limited at least in the current format, in light of what is already known in the literature.

Pali1-mediated PRC2 activation was previously shown by Conway et al. (Mol Cell. 2018). In the same study, loss of Pali1 was reported to reduce H3K27me3

enrichment on PRC2 targets. Data from the current study further indicated that Pali1 could contribute to PRC2 activation via two distinct mechanisms, including enhanced nucleosome binding as well as allosteric activation through Eed. The latter mechanism required direct interaction of methyl Pali1 with Eed. The structural study on Eed in complex with methylated Pali1 peptides revealed structures very similar to that of Eed bound to a methylated Jarid2 peptide, which was reported a few years ago by Sanulli et al. (Mol Cell. 2015). In that earlier paper, methylated Jarid2 was also indicated to regulate cell differentiation based on a series of in vitro methylation assays and ChIP analysis. Pali1 was not the first non-histone substrate ever identified. In addition to Jarid2 mentioned above, Ezh2 and Elongin A were both recently found to be methylated by PRC2 to regulate H3K27 methylation or transcription.

Although the authors observed that overexpression of the Pali1-K1241A mutant in K562 cells failed to promote differentiation as compared with wildtype Pali1, the mechanism was completely lacking. Was the H3K27me3 level changed on chromatin loci implicated in the regulation of differentiation? Was such change if any caused by the lack of allosteric activation of PRC2?

Publication is not recommended unless major revision can be completed to add novelty and to provide mechanistic insights on the functional study.

We now added data showing that PALI1 facilitates nucleosome binding by PRC2 and strengthened our data already demonstrated DNA binding. Hence, while the mechanism for subunit-induced allosteric activation was indeed demonstrated for JARID2, the new data herein is now provide the first molecular mechanism for PALI1-mediated regulation of PRC2: PALI1 facilitate DNA binding (Fig 6), nucleosome binding (Fig 7) and, independently, allosterically activate PRC2 (Fig 2-4); this mechanism is novel: while Conway et al 2018 show that PALI1 enhances the HMTase activity of PRC2, our work now identifies how this is done. As this is the first work to provide the molecular basis for PALI1-mediated regulation of PRC2, we now also modified the title to explicitly state the mechanism that we discovered: "*PALI1 facilitates DNA and nucleosome binding by PRC2 and triggers an allosteric activation of catalysis*".

Major points:

Throughout the manuscript, the authors used the truncated PRC2-Pali1PIR complex for the in vitro methylation and DNA/nucleosome binding assays. Some of the major conclusions need to be confirmed with full-length PRC2-Pali1 complex.

Response to point 1: We now included also HMTase activity assays using a complex of PRC2 with the full-length PALI1 (PRC2-PALI1_{FL}; Supplementary Fig. 1d, e, f). As expected, the PRC2-PALI1_{FL} complex is methylated by PRC2 in vitro and observed with higher HMTase activity than the PRC2 core complex, although the full-length PALI1 protein (PALI1_{FL}) is not as stable as the PALI1_{PIR}.

In Fig. 4b, it is unclear why the authors chose to use a mixture of tailed mononucleosomes and free DNA for EMSA. In addition, what is the binding affinity between PRC2-Pali1 and the nucleosome core without the tail? It is difficult to quantitate the contribution of the nucleosome core to the binding when the DNA tail is present.

Response to point 2: We now added quantitative binding assays using fluorescently labelled nucleosome core particles without linker DNA or free DNA (Fig. 7a, b). The results show that PALI1 increases the affinity of PRC2 for nucleosome core particles (147 bp DNA) by >50 fold, indicating that PALI1 is a nucleosome-binding subunit of PRC2.

Is there experimental evidence to show that Pali1 methylation occurs prior to H3K27 methylation in cells? Otherwise, what would be the functional importance for methyl Pali1-mediated PRC2 activation when existing H3K27me3 can also activate PRC2?

Response to point 3: We now added data showing that PRC2-induced methylation of PALI1 is enhanced in the presence of nucleosome substrates in vitro (Supplementary Fig 2h). It implies that the methylation of PAL1 is more prominent when the PRC2-PALI1 complex is bound to chromatin. Further testing this in cells and placing it in the context of transcriptional regulation would require significant development, as so far we were unable to raise antibodies against the methylated form of PALI1. To our knowledge, there is also no experimental evidence that PRC2-induced methylation of endogenously-expressed JARID2 occurs before (or simultaneously with) H3K27 methylation in cells. This is indeed an important question, but it is beyond the scope of this study.

The authors used an overexpression system to compare the effect of wildtype Pali1 and the Pali1-K1241A mutant on K562 cell differentiation. The authors showed roughly comparable levels of Pali1 expression by Western blot. However, it is unclear how sensitive the observed cell phenotype is to the protein expression level. If even a subtle difference in protein expression may cause a dramatic change of the phenotype, the current experiment needs to be better calibrated.

Response to point 4: All the proteins of interest are expressed using a polycistronic lentivirus construct together with an EGFP, and flow cytometry is used to control for the expression level before experiments are starting. Nevertheless, we now added an experiment where flow cytometry gating was applied to select cells based on the different expression level of PALI1. The results show that the phenotype of CD

expression level is insensitive to the ectopic expression level of PALI1 (Supplementary Fig 9c). We also show that the knockout of EED leads to the depletion of the ectopically expressed PALI1 protein (Supplementary Fig 9f), implying that the protein level of PALI1 in the cell is limited by the amount of PRC2 in the cells, not the expression level of the PALI1 mRNA. We now also carried out all the flow cytometry experiments after replacing the CD71 antibody from the original submission with a CD44 antibody, which is a more commonly studied CD marker and we found it as more robust (Fig 8d and Supplementary Fig 9a-e).

The present cell proliferation and differentiation marker expression data are insufficient to conclude that methyl Pali1-mediated PRC2 activation has an important function in cell differentiation. Given that the global H3K27me3 level stayed unchanged for cells expressing Pali1-K1241A, it is necessary to investigate by ChIP analysis whether the H3K27me3 level on specific PRC2 targets is affected by this mutant. Chromatin targeting of PRC2-Pali1 needs to be studied in parallel as a control.

Response to point 5: The target genes of PALI1 in cells were not identified so far, with at least one (excellent) lab that has reported their inability to ChIP for PALI1 (Conway et al Mol Cell. 2018;70(3):408-421.e8.). In agreement with Conway et al, all the attempts that we did so far to identify direct target genes of PALI1 in K562 cells using ChIP or CUT&Tag were unsuccessful. This makes it impossible for us to detect changes of the H3K27me3 mark in PALI1-target genes. We now added to the manuscript ChIP-qPCR with the overexpression of PALI1 in HEK293T cells (Supplementary Fig 10). Since we do not know what are the target genes of endogenous PALI1 in HEK293T cells, we selected a few known PRC2 target genes. Yet, although we could detect the recruitment of PALI1 to chromatin there, we could not detect significant changes in H3K27me3 or the recruitment of EZH2 (Supplementary Fig 10). This is possibly the result of redundancy between accessory subunits and/or an insufficient amount of PCL proteins at these target genes. PCL proteins bind to PRC2 together with PALI1 (Hauri et al. Cell Rep. 2016;17(2):583-595.) and are required for the targeting of PRC2 to CpG islands in cells (Li et al. Nature. 2017;549(7671):287-291.). Conway et al knocked out PALI1 in mESCs but observed only a modest reduction of the H3K27me3 mark (typically 20%-30%; see Fig 6A in Conway et al Mol Cell. 2018;70(3):408-421.e8.). Conway et al observed a larger reduction of H3K27me3 only when additional accessory subunits were depleted (see Fig 6 in Conway et al). In the case of the PALI1 K1241A mutation, the effect would be expected to be even smaller than what seen for a PALI1 knockout, as the K1241A mutation does not completely "kill" the HMTase activity of PALI1 but just prevent an allosteric activation (see Fig 2A in our manuscript). Hence, while it is indeed important to follow up on the phenotype of the K1241A mutation in vivo, it would require significant development of experimental systems beyond this study. Therefore, we now moderated statements that we made in the original submission regarding the involvement of PALI1 in cell differentiation: a sentence deleted from L81, the discussion at L367-382 into the CD

markers quantification was revised and Fig 8e was revised. We also added a sentence reminding the reader that PCL proteins might also be involved in the recruitment of PRC2-PALI1 to chromatin in cells (L292-294).

Reviewer #3 (Remarks to the Author):

Polycomb repressive complex 2 (PRC2) is a histone methyltransferase that is essential for maintaining cellular identity during development and can be divided into two major classes, PRC2.1 and PRC2.2. PRC2.2 contains the accessory subunit, JARID2 which is known to trigger an allosteric activation of histone methyltransferase. However, it is not known which specific subunits of PRC2.1 is the counterpart of JARID2 to facilitate PRC2 activity. This manuscript explores the roles and molecular mechanisms of the PRC2.1 specific subunit, PALI1 in an allosteric activation of PRC2.1. Authors indicate that the PRC2-binding domain of PALI1 enhances H3K27 methyltransferase activity of PRC2.1 by two independent mechanisms. (1) DNA- and substrate binding and (2) allosteric stimulation. The mechanistic resemblance between PALI1 and JARID2 indicates both proteins regulate allosterically activation of PRC2.1 and PRC2.2 complex respectively. Although most of experiments are done carefully and the

conclusions are well supported by the results, in vivo relevance of the findings is limited. I do have the following comments which I hope will improve the impact of the study:

Major points

1) It is unclear to me regarding complexity of PRC2.1. PRC2.1 contains one of PCL proteins together with either EPOP or PALI1. If EPOP and PALI1 form a complex with PRC2 in a mutually manner, how does EPOP-containing PRC2 achieve the stimulation of PRC2? In line with this aspect, does PALI2 have a similar property to PALI1 in terms of conserved Lysine residues that potentially undergo methylation and stimulation of DNA binding and activity of PRC2? Also, If PALI1 facilitates chromatin binding of PRC2, what is the function of PCL proteins? Authors should discuss in the manuscript.

EPOP: We did not identify lysins in EPOP that are consistently methylated in cells (Fig. 1). Therefore, we do not have a basis to consider EPOP as an allosteric activator of PRC2, in agreement with EPOP being a negative regulator of PRC2 in vivo (Beringer et al. Mol Cell. 2016;64(4):645-658.; Liefke et al Mol Cell. 2016;64(4):659-672.). We now added a discussion into this in L245-248.

PALI2: We now added new experiments showing that PALI2 K1558 and its adjacent phenylalanine are homologous to PALI1 K1241 (Fig 5a) and in its trimethyl form it binds to EED (Fig 5b) and triggers an allosteric activation of PRC2 (Fig 5c). This data is discussed in a new results section dedicated to PALI2 (L244-L263).

PCL proteins: Our data indicate that PALI1 binds to DNA without apparent sequence specificity. It is, therefore, possible that interactions with PCL proteins help to direct PALI1 to its target sites on chromatin. This is now discussed in L292-294.

2) Authors convincingly demonstrated that PALI1 K1241 is methylated in vitro and in vivo, and that this methylation facilitates DNA binding of PRC2 and the methyltransferase activity toward H3K27. However, it is unclear about (1) the timing and (2) the spatial constraint of methylation of PALI1 in vivo. If the antibody against PALI1 K1241me_{2/3} or at least the antibody against methyl-lysines are available, authors should perform a biochemical purification of PRC2 from the fractionated cell lysates (cytosol, Nuclear, and chromatin bound) and perform western blotting to assess which fractions are enriched with methylated PALI1.

We now included an additional experiment, demonstrating that the methylation of PALI1 by PRC2 is more efficient in the presence of nucleosome substrates in vitro (Supplementary Fig. 2h). Despite multiple attempts, we were so far unsuccessful in raising antibodies against the methylated form of PALI1. We also were unable to perform biochemical purification of PRC2 following immunoblotting using an anti-methyl-lysine antibody, possibly because of the high level of purity that is required to obtain interpretable immunoblotting results while using such antibody of broad specificity. While our in vitro results are now implying that PALI1 is more likely to be methylated upon binding to chromatin (Supplementary Fig. 2h), following up on this in vivo would require a significant development beyond this manuscript.

The following experiment should be carried out:

3) If the authors propose a functional resemblance between PALI1 to JARID2, I would suggest to assess the importance of the +1 adjacent aromatic residue (PALI1 F1242A) in both EED binding and the stimulation of PRC2.

We now performed the requested peptide binding and HMTase assays (Fig. 4c-e). In agreement with other experiments in the manuscript, the results indicate that an F1242A mutation in the PALI1-K1241me₃ peptide prevents it from binding to EED (Fig 4c) and does not allow the stimulation of PRC2 (Fig 4d, e).

4) In Figure 5C and 5d, authors should perform H3K27me3 and PRC2 ChIP-qPCR at couple of the PRC2 target loci in the erythroid lineage to investigate that de novo targeting of PRC2 and the H3K27me3 deposition to a target gene is facilitated by PALI1 WT but not PALI1 K1241A.

See our response to the last point of reviewer 2.

Minor comments

1) I think the title of this manuscript is a little bit overstated. While the authors revealed the functional similarities between LARID2 and PALI1, they did not show the evidence of convergent evolution between them.

We now changed the title to “PALI1 facilitates DNA and nucleosome binding by PRC2 and triggers an allosteric activation of catalysis”. We believe that the new title is directly supported by experiments in the revised manuscript.

2) Related to Fig.1, the authors should show if PALI1 is methylated by PRC2 in vivo.

As explained above in our response to Major point 2, despite multiple attempts we were unable to produce an antibody for the methylated form of PALI1. Therefore, we could not use immunoblotting to show that PRC2 is the methyltransferase in cells. The alternative approach of quantitative proteomics would require data of better quality than most of the publicly available AP-MS datasets that we used to generate Fig 1 and would, therefore, require a significant development beyond this study.

3) In Figure 4b, authors argued that PALI1^{PIR} facilitates DNA binding by PRC2. While, in the manuscript at line 263-266, it is described that the probe can detect interaction at either the centered or the off-centered position, or the free DNA, the conclusion from this experiment is that the PRC2^{PIP} increased the affinity of PRC2 to nucleosome. How about the binding of the free DNA and either the center or off-center?

We now included quantitative binding assays with fluorescently labelled nucleosome core particles with and without a DNA linker (Fig 7a, b). The results show that PALI1 increases the affinity of PRC2 to mono-nucleosomes by >50-fold, even in the absence of a linker DNA (top gels in Fig 7a and continues line in Fig 7b). Adding a linker DNA further increases the affinity by approximately 3-fold (bottom gels in Fig 7a and dashed line in Fig 7b). The new assays supporting interactions with nucleosomes are now in a separate chapter titled “PALI1 facilitates nucleosome binding by PRC2” (L295-L338). Binding assays with only DNA as the probe (i.e. without nucleosomes) are presented in Fig 6a, showing that PALI1^{PIR} increases the affinity of PRC2 to DNA. Collectively, these assays directly show that PALI1 facilitates DNA and nucleosome binding by PRC2.

4) Related to line 286, the authors should discuss the current knowledge of the expression pattern and the prognostic role of PALI1 in myeloid malignancies.

The deletion of the *LCOR* (i.e. PALI1) and *JARID2* loci in hematopoietic malignancies is now discussed in L345-351.

5) In Figure 5 where the authors discussed the biological significance of enhancement of PRC2 catalytic activities by PALI1, the experiments related to PALI1 knock out should be included.

We now discuss the biological significance of PALI1 as reflected from knockout experiments in mouse, carried out by Conway et al 2018 in L377-L379.

REVIEWERS' COMMENTS

Reviewer #1 (Remarks to the Author):

The authors addressed all my major criticisms adequately.

Reviewer #2 (Remarks to the Author):

The authors have adequately addressed most of the concerns except the one below.

They attempted to characterize PRC2-Pali1-FL complex in Supplementary Fig. 1d, e, f. However, some of these new data are problematic:

- (1) Coomassie blue band corresponding to Pali1-FL is not visible at all in Supplementary Fig. 1d.
- (2) SDS-PAGE gel should be provided for the FPLC fractions in Supplementary Fig. 1e, to show the elution profile corresponds to an intact PRC2-Pali1-FL complex.

Reviewer #3 (Remarks to the Author):

The authors have thoroughly addressed most of comments from reviewers including myself and added new data that strengthen the manuscript.

We wish to thank the reviewers for dedicating their time to assess our revised manuscript. We addressed below the additional points made by reviewer 2 (in blue).

REVIEWERS' COMMENTS

Reviewer #1 (Remarks to the Author):

The authors addressed all my major criticisms adequately.

Reviewer #2 (Remarks to the Author):

The authors have adequately addressed most of the concerns except the one below.

They attempted to characterize PRC2-Pali1-FL complex in Supplementary Fig. 1d, e, f. However, some of these new data are problematic:

- (1) Coomassie blue band corresponding to Pali1-FL is not visible at all in Supplementary Fig. 1d.
- (2) SDS-PAGE gel should be provided for the FPLC fractions in Supplementary Fig. 1e, to show the elution profile corresponds to an intact PRC2-Pali1-FL complex.

We now added to Supplementary Fig 1e the Coomassie-stained SDS-PAGE of the FPLC fractions collected during the purification of the PRC2-PALI1_{FL} construct. We wish to reiterate, as we explained in our response to the last round of review, that the full-length human PALI1 protein construct (PALI1_{FL}) is unstable and, for that reason, we did not include experiments with that protein in the original submission but rather added them during the revision after we were explicitly requested to do so. During the previous revision cycle we also wrote it in the *Results* section that “...the full-length PALI1 (PALI1_{FL}) was not as stable as the PALI1_{PIR} construct (Supplementary Fig. 1d, e)...”. Specifically, PALI1_{FL} is obtained in multiple truncated forms and, therefore, was not resolved on the Coomassie-stained gels presented in Supplementary Fig. 1d. Nevertheless, the multiple truncated forms of the PALI1_{FL} construct can be seen on the radiogram of supplementary Fig 1f, likely because of the strong intensity of the radioactive signal. Supplementary Figure 1f also shows that PALI1_{FL} is active in stimulating PRC2.

Reviewer #3 (Remarks to the Author):

The authors have thoroughly addressed most of comments from reviewers including myself and added new data that strengthen the manuscript.